# Large Scale Learning on Non-Homophilous Graphs: New Benchmarks and Strong Simple Methods

**Derek Lim***
Cornell University
dl772@cornell.edu

**Felix Hohne***
Cornell University
fmh42@cornell.edu

**Xiuyu Li***
Cornell University
xl289@cornell.edu

**Sijia Linda Huang**
Cornell University
sh837@cornell.edu

**Vaishnavi Gupta**
Cornell University
vg222@cornell.edu

**Omkar Bhalerao**
Cornell University
opb7@cornell.edu

**Ser-Nam Lim**
Facebook AI
sernam@gmail.com

## Abstract

Many widely used datasets for graph machine learning tasks have generally been homophilous, where nodes with similar labels connect to each other. Recently, new Graph Neural Networks (GNNs) have been developed that move beyond the homophily regime; however, their evaluation has often been conducted on small graphs with limited application domains. We collect and introduce diverse non-homophilous datasets from a variety of application areas that have up to 384x more nodes and 1398x more edges than prior datasets. We further show that existing scalable graph learning and graph minibatching techniques lead to performance degradation on these non-homophilous datasets, thus highlighting the need for further work on scalable non-homophilous methods. To address these concerns, we introduce LINKX — a strong simple method that admits straightforward minibatch training and inference. Extensive experimental results with representative simple methods and GNNs across our proposed datasets show that LINKX achieves state-of-the-art performance for learning on non-homophilous graphs. Our codes and data are available at https://github.com/CUAI/Non-Homophily-Large-Scale.

## 1 Introduction

Graph learning methods generate predictions by leveraging complex inductive biases captured in the topology of the graph [7]. A large volume of work in this area, including graph neural networks (GNNs), exploits *homophily* as a strong inductive bias, where connected nodes tend to be similar to each other in terms of labels [46, 3]. Such assumptions of homophily, however, do not always hold true. For example, malicious node detection, a key application of graph machine learning, is known to be non-homophilous in many settings [55, 13, 25, 11].

Further, while new GNNs that work better in these non-homophilous settings have been developed [82, 44, 81, 17, 15, 73, 36, 35, 9, 54], their evaluation is limited to a few graph datasets used by Pei et al. [58] (collected by [61, 66, 48]) that have certain undesirable properties such as small size, narrow range of application areas, and high variance between different train/test splits [82]. Consequently, method scalability has not been thoroughly studied in non-homophilous graph learning. In fact, many non-homophilous techniques frequently require more parameters and computational resources [82, 1, 17], which is neither evident nor detrimental when they are evaluated on very small datasets. Even though scalable graph learning techniques do exist, these methods generally cannot

---

*Equal contribution.

35th Conference on Neural Information Processing Systems (NeurIPS 2021).

be directly applied to the non-homophilous setting, as they oftentimes assume homophily in their construction [71, 32, 20, 10].

Non-homophily in graphs also degrades proven graph learning techniques that have been instrumental to strong performance in scalable graph learning. For instance, label propagation, personalized PageRank, and low-pass graph filtering have been used for scalable graph representation learning models, but these methods all assume homophily [71, 32, 20, 10]. Moreover, we give empirical evidence that existing minibatching techniques in graph learning [16, 77] significantly degrade performance in non-homophilous settings. In response, we develop a novel model, LINKX, that addresses these concerns; LINKX outperforms existing graph learning methods on large-scale non-homophilous datasets and admits a simple minibatching procedure that maintains strong performance.

To summarize, we demonstrate three key areas of deficiency as mentioned above, namely: (1) that there is a lack of large, high-quality datasets covering different non-homophilous applications, (2) that current graph minibatching techniques and scalable methods do not work well in non-homophilous settings, and (3) that prior non-homophilous methods are not scalable. To these ends, this paper makes the following contributions:

**Dataset Collection and Benchmarking.** We collect a diverse series of large, non-homophilous graph datasets and define new node features and tasks for classification. These datasets are substantially larger than previous non-homophilous datasets, span wider application areas, and capture different types of complex label-topology relationships. With these proposed datasets, we conduct extensive experiments with 14 graph learning methods and 3 graph minibatching techniques that are broadly representative of the graph machine learning model space.

**Analyzing Scalable Methods and Minibatching.** We analyze current graph minibatching techniques like GraphSAINT [77] in non-homophilous settings, showing that they substantially degrade performance in experiments. Also, we show empirically that scalable methods for graph learning like SGC and C&S [71, 32] do not perform well in non-homophilous settings — even though they achieve state-of-the-art results on many homophilous graph benchmarks. Finally, we demonstrate that existing non-homophilous methods often suffer from issues with scalability and performance in large non-homophilous graphs, in large part due to a lack of study of large-scale non-homophilous graph learning.

**LINKX: a strong, simple method.** We propose a simple method LINKX that achieves excellent results for non-homophilous graphs while overcoming the above-mentioned minibatching issues. LINKX works by separately embedding the adjacency $\mathbf{A}$ and node features $\mathbf{X}$, then combining them with multilayer perceptrons and simple transformations, as illustrated in Figure 1. It generalizes node feature MLP and LINK regression [79], two baselines that often work well on non-homophilous graphs. This method is simple to train and evaluate in a minibatched fashion, and does not face the performance degradation that other methods do in the minibatch setting. We develop the model and give more details in Section 4.

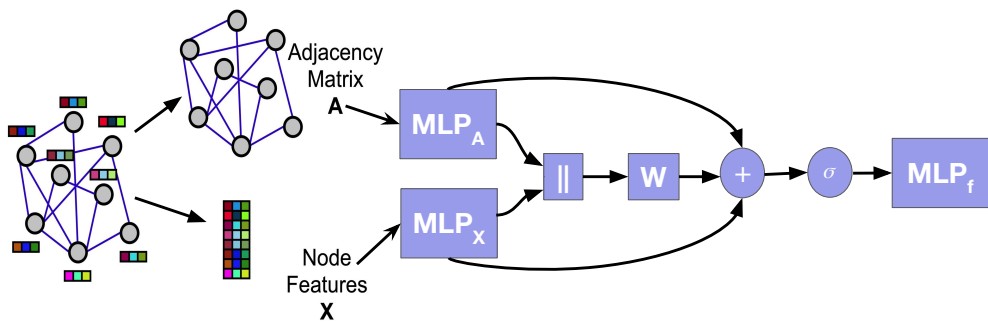

Figure 1: Our model LINKX separately embeds node features and adjacency information with MLPs, combines the embeddings together by concatenation, then uses a final MLP to generate predictions.

## 2 Prior Work

**Graph Representation Learning.** Graph neural networks [28, 38, 69] have demonstrated their utility on a variety of graph machine learning tasks. Most GNNs are constructed by stacking layers that propagate transformed node features, which are then aggregated via different mechanisms. The neighborhood aggregation used in many existing GNNs implicitly leverage homophily, so they often fail to generalize on non-homophilous graphs [82, 6]. Indeed, a wide range of GNNs operate as low-pass graph filters [53, 71, 6] that smooth features over the graph topology, which produces similar representations and thus similar predictions for neighboring nodes.

**Scalable methods.** A variety of scalable graph learning methods have been developed for efficient computation in larger datasets [77, 16, 75, 28, 71, 32, 20, 10]. Many of these methods explicitly make use of an assumption of homophily in the data [71, 32, 20, 10]. By leveraging this assumption, several simple, inexpensive models are able to achieve state-of-the-art performance on homophilic datasets [71, 32]. However, these methods are unable to achieve comparable performance in non-homophilous settings, as we show empirically in Section 5.

**Graph sampling.** As node representations depend on other nodes in the graph, there are no simple minibatching techniques in graph learning as there are for i.i.d. data. To scale to large graphs, one line of work samples nodes that are used in each layer of a graph neural network [28, 75, 14]. Another family of methods samples subgraphs of an input graph, then passes each subgraph through a GNN to make a prediction for each node of the subgraph [16, 76, 77]. While these methods are useful for scalable graph learning, we show that they substantially degrade performance in our non-homphilous experiments (see Section 5).

**Non-Homophilous methods.** Various GNNs have been proposed to achieve higher performance in low-homophily settings [82, 44, 81, 17, 15, 73, 36, 35]. Geom-GCN [58] introduces a geometric aggregation scheme, MixHop [1] proposes a graph convolutional layer that mixes powers of the adjacency matrix, GPR-GNN [17] features learnable weights that can be positive and negative in feature propagation, GCNII [15] allows deep graph convolutional networks with relieved oversmoothing, which empirically performs better in non-homophilous settings, and $H_2$GCN [82] shows that separation of ego and neighbor embeddings, aggregation in higher-order neighborhoods, and the combination of intermediate representations improves GNN performance in low-homophily.

There are several recurring design decisions across these methods that appear to strengthen performance in non-homophilous settings: using higher-order neighborhoods, decoupling neighbor information from ego information, and combining graph information at different scales [82]. Many of these design choices require additional overhead (see Section 4.3), thus reducing their scalability.

**Datasets.** The widely used citation networks Cora, Citeseer, and Pubmed [62, 74] are highly homophilous (see Appendix A) [82]. Recently, the Open Graph Benchmark [31] has provided a series of datasets and leaderboards that improve the quality of evaluation in graph representation learning; however, most of the node classification datasets tend to be homophilous, as noted in past work [82] and expanded upon in Appendix A.2. A comparable set of high-quality benchmarks to evaluate non-homophilous methods does not currently exist.

## 3 Datasets for Non-Homophilous Graph Learning

### 3.1 Currently Used Datasets

The most widely used datasets to evaluate non-homophilous graph representation learning methods were used by Pei et al. [58] (and collected by [61, 66, 48]); see our Table 1 for statistics. However, these datasets have fundamental issues. First, they are very small — the Cornell, Texas, and Wisconsin datasets have between 180-250 nodes, and the largest dataset Actor has 7,600 nodes. In analogy to certain pitfalls of graph neural network evaluation on small (homophilic) datasets discussed in [63], evaluation on the datasets of Pei et al. [58] is plagued by high variance across different train/test splits (see results in [82]). The small size of these datasets may tend to create models that are more prone to overfitting [21], which prevents the scaling up of GNNs designed for non-homophilous settings.

Peel [57] also studies node classification on network datasets with various types of relationships between edges and labels. However, they only study methods that act on graph topology, and thus their datasets do not necessarily have node features. We take inspiration from their work, by

testing on Pokec and Facebook networks with node features that we define, and by introducing other year-prediction tasks on citation networks that have node features.

## 3.2 An Improved Homophily Measure

Various metrics have been proposed to measure the homophily of a graph. However, these metrics are sensitive to the number of classes and the number of nodes in each class. Let $G = (V, E)$ be a graph with $n$ nodes, none of which are isolated. Further let each node $u \in V$ have a class label $k_u \in \{0, 1, \ldots, C - 1\}$ for some number of classes $C$, and denote by $C_k$ the set of nodes in class $k$. The edge homophily [82] is the proportion of edges that connect two nodes of the same class:

$$h = \frac{|\{(u, v) \in E : k_u = k_v\}|}{|E|}. \tag{1}$$

Another related measure is what we call the node homophily [58], defined as $\frac{1}{|V|} \sum_{u \in V} \frac{d_u^{(k_u)}}{d_u}$, in which $d_u$ is the number of neighbors of node $u$, and $d_u^{(k_u)}$ is the number of neighbors of $u$ that have the same class label. We focus on the edge homophily (1) in this work, but find that node homophily tends to have similar qualitative behavior in experiments.

The sensitivity of edge homophily to the number of classes and size of each class limits its utility. We consider a null model for graphs in which the graph topology is independent of the labels; suppose that nodes with corresponding labels are fixed, and include edges uniformly at random in the graph that are independent of node labels. Under this null model, a node $u \in V$ would be expected to have $d_u^{(k_u)}/d_u \approx |C_{k_u}|/n$ as the proportion of nodes of the same class that they connect to [3]. For a dataset with $C$ balanced classes, we would thus expect the edge homophily to be around $\frac{1}{C}$, so the interpretation of the measure depends on the number of classes. Also, if classes are imbalanced, then the edge homophily may be misleadingly large. For instance, if 99% of nodes were of one class, then most edges would likely be within that same class, so the edge homophily would be high, even when the graph is generated from the null model where labels are independent of graph topology. Thus, the edge homophily does not capture deviation of the label distribution from the null model.

We introduce a metric that better captures the presence or absence of homophily. Unlike the edge homophily, our metric measures excess homophily that is not expected from the above null model where edges are randomly wired. Our metric does not distinguish between different non-homophilous settings (such as heterophily or independent edges); we believe that there are too many degrees of freedom in non-homophilous settings for a single scalar quantity to be able to distinguish them all. Our measure is given as:

$$\hat{h} = \frac{1}{C - 1} \sum_{k=0}^{C-1} \left[ h_k - \frac{|C_k|}{n} \right]_+, \tag{2}$$

where $[a]_+ = \max(a, 0)$, and $h_k$ is the class-wise homophily metric

$$h_k = \frac{\sum_{u \in C_k} d_u^{(k_u)}}{\sum_{u \in C_k} d_u}. \tag{3}$$

Note that $\hat{h} \in [0, 1]$, with a fully homophilous graph (in which every node is only connected to nodes of the same class) having $\hat{h} = 1$. Since each class-wise homophily metric $h_k$ only contributes positive deviations from the null expected proportion $|C_k|/n$, the class-imbalance problem is substantially mitigated. Also, graphs in which edges are independent of node labels are expected to have $\hat{h} \approx 0$, for any number of classes. Our measure $\hat{h}$ measures presence of homophily, but does not distinguish between the many types of possibly non-homophilous relationships. This is reasonable given the diversity of non-homophilous relationships. For example, non-homophily can imply independence of edges and classes, extreme heterophily, connections only among subsets of classes, or certain chemically / biologically determined relationships. Indeed, these relationships are very different, and are better captured by more than one scalar quantity, such as the compatibility matrices presented in the appendix. Further discussion is given in Appendix A.

Table 1: Statistics for previously used datasets from Pei et al. [58] (collected by [61, 66, 48]). #C is the number of node classes. The highest number of nodes or edges overall are bolded.

| Dataset | # Nodes | # Edges | # Feat. | # C | Context | Edge hom. | $\hat{h}$ (ours) |
|---|---|---|---|---|---|---|---|
| Chameleon | 2,277 | 36,101 | 2,325 | 5 | Wiki pages | .23 | .062 |
| Cornell | 183 | 295 | 1,703 | 5 | Web pages | .30 | .047 |
| Actor | **7,600** | 29,926 | 931 | 5 | Actors in movies | .22 | .011 |
| Squirrel | 5,201 | **216,933** | 2,089 | 5 | Wiki pages | .22 | .025 |
| Texas | 183 | 309 | 1,703 | 5 | Web pages | .11 | .001 |
| Wisconsin | 251 | 499 | 1,703 | 5 | Web pages | .21 | .094 |

## 3.3 Proposed Datasets

Here, we detail the non-homophilous datasets that we propose for graph machine learning evaluation. Our datasets and tasks span diverse application areas. **Penn94** [67], **Pokec** [41], **genius** [43], and **twitch-gamers** [60] are online social networks, where the task is to predict reported gender, certain account labels, or use of explicit content on user accounts. For the citation networks **arXiv-year** [31] and **snap-patents** [42, 41] the goal is to predict year of paper publication or the year that a patent is granted. The dataset **wiki** consists of Wikipedia articles, where the goal is to predict total page views of each article. Detailed descriptions about the graph structure, node features, node labels, and licenses of each dataset are given in Appendix D.2.

Most of these datasets have been used for evaluation of graph machine learning models in past work; we make adjustments such as modifying node labels and adding node features that allow for evaluation of GNNs in non-homophilous settings. We define node features for Pokec, genius, and snap-patents, and we also define node labels for arXiv-year, snap-patents, and genius. Additionally, we crawl and clean the large-scale wiki dataset — a new Wikipedia dataset where the task is to predict page views, which is non-homophilous with respect to the graph of articles connected by links between articles (see Appendix D.3). This wiki dataset has 1,925,342 nodes and 303,434,860 edges, so training and inference require scalable algorithms.

Basic dataset statistics are given in Table 2. Note the substantial difference between the size of our datasets and those of Pei et al. [58] in Table 1; our datasets have up to 384x more nodes and 1398x more edges. The homophily measures along with the lower empirical performance of homophily-assuming models (Section 5) and examination of compatibility matrices (Appendix A) show that our datasets are indeed non-homophilous. As there is little study in large-scale non-homophilous graph learning, our proposed large datasets strongly motivate the need for developing a new, scalable approach that can accurately learn on non-homophilous graphs.

Table 2: Statistics of our proposed non-homophilous graph datasets. # C is the number of distinct node classes. Note that our datasets come from more diverse applications areas and are much larger than those shown in Table 1, with up to 384x more nodes and 1398x more edges.

| Dataset | # Nodes | # Edges | # Feat. | # C | Class types | Edge hom. | $\hat{h}$ (ours) |
|---|---|---|---|---|---|---|---|
| Penn94 | 41,554 | 1,362,229 | 5 | 2 | gender | .470 | .046 |
| pokec | 1,632,803 | 30,622,564 | 65 | 2 | gender | .445 | .000 |
| arXiv-year | 169,343 | 1,166,243 | 128 | 5 | pub year | .222 | .272 |
| snap-patents | **2,923,922** | 13,975,788 | 269 | 5 | time granted | .073 | .100 |
| genius | 421,961 | 984,979 | 12 | 2 | marked act. | .618 | .080 |
| twitch-gamers | 168,114 | 6,797,557 | 7 | 2 | mature content | .545 | .090 |
| wiki | 1,925,342 | **303,434,860** | 600 | 5 | views | .389 | .107 |

## 4 LINKX: A New Scalable Model

In this section, we introduce our novel model, LINKX, for scalable node classification in non-homophilous settings. LINKX is built out of multilayer perceptrons (MLPs) and linear transformations, thus making it simple and scalable. It also admits simple row-wise minibatching procedures that allow it to perform well on large non-homophilous graphs. As a result, LINKX is able to

circumvent aforementioned issues of graph minibatching and non-homophilous GNNs in large-scale non-homophilous settings.

## 4.1 Motivation from two simple baselines

Here, we detail two simple baselines for node classification that we build on to develop LINKX.

**MLP on node features.** A naïve method for node classification is to ignore the graph topology and simply train an MLP on node features. For the same reason that the graph topology has more complicated relationships with label distributions in non-homophilous graphs, many GNNs are not able to effectively leverage the graph topology in these settings. Thus, MLPs can actually perform comparatively well on non-homophilous graphs — achieving higher or approximately equal performance to various GNNs [82].

**LINK regression on graph topology.** On the other extreme, there is LINK [79] — a simple baseline that only utilizes graph topology. In particular, we consider LINK regression, which trains a logistic regression model in which each node's features are taken from a column of the adjacency matrix. Letting $\mathbf{A} \in \{0,1\}^{n \times n}$ be the binary adjacency matrix of the graph, and $\mathbf{W} \in \mathbb{R}^{c \times n}$ be a learned weight matrix, LINK computes class probabilities as

$$\mathbf{Y} = \mathrm{softmax}(\mathbf{WA}). \tag{4}$$

Let $u \in \{1, \ldots, n\}$ be a specific node, and let $k \in \{1, \ldots, c\}$ be a specific class. Then, expanding the matrix multiplication, the log-odds of node $u$ belonging to class $k$ is given by

$$(\mathbf{WA})_{ku} = \sum_{v \in \mathcal{N}(u)} \mathbf{W}_{kv}, \tag{5}$$

where $\mathcal{N}(u)$ contains the 1-hop neighbors of $u$. In other words, the logit is given by the sum of weights $\mathbf{W}_{kv}$ across the 1-hop neighbors of $u$. If a specific node $v$ has many neighbors of class $k$, then $\mathbf{W}_{kv}$ is probably large, as we would expect with a high probability that any neighbor of $v$ is of class $k$. In this sense, LINK is like a 2-hop method: for a given node $u$, the probability of being in a given class is related to the class memberships of $u$'s 2-hop neighbors in $\mathcal{N}(v)$ for each neighbor $v \in \mathcal{N}(u)$. Related interpretations of LINK as a method acting on 2-hop paths between nodes are given by Altenburger and Ugander [3].

Though it is simple and has been overlooked in the recent non-homophilous GNN literature, LINK has been found to perform well in certain node classification tasks like gender prediction in social networks [3, 4]. A major reason why LINK does well in many settings is exactly because it acts as a 2-hop method. For example, while 1-hop neighbors are often not so informative for gender prediction in social networks due to lack of homophily, 2-hop neighbors are very informative due to so-called "monophily," whereby many nodes have extreme preferences for connecting to a certain class [3]. Beyond just gender prediction, we show in Section 5 that LINK empirically outperforms many models across the various application areas of the non-homophilous datasets we propose.

## 4.2 LINKX

We combine these two simple baselines through simple linear transformations and component-wise nonlinearities. Let $\mathbf{X} \in \mathbb{R}^{D \times n}$ denote the matrix of node features with input dimension $D$, and let $[\mathbf{h}_1; \mathbf{h}_2]$ denote concatenation of vectors $\mathbf{h}_1$ and $\mathbf{h}_2$. Then our model outputs predictions $\mathbf{Y}$ through the following mapping:

$$\mathbf{h_A} = \mathrm{MLP_A}(\mathbf{A}) \in \mathbb{R}^{d \times n} \tag{6}$$

$$\mathbf{h_X} = \mathrm{MLP_X}(\mathbf{X}) \in \mathbb{R}^{d \times n} \tag{7}$$

$$\mathbf{Y} = \mathrm{MLP}_f \left( \sigma \left( \mathbf{W}[\mathbf{h_A}; \mathbf{h_X}] + \mathbf{h_A} + \mathbf{h_X} \right) \right), \tag{8}$$

in which $d$ is the hidden dimension, $\mathbf{W} \in \mathbb{R}^{d \times 2d}$ is a weight matrix, and $\sigma$ is a component-wise nonlinearity (which we take to be ReLU). We call our model LINKX, as it extends LINK with node feature information from the matrix $\mathbf{X}$. A diagram of LINKX is given in Figure 1.

First, LINKX computes hidden representations $\mathbf{h_A}$ of the adjacency (extending LINK) and $\mathbf{h_X}$ of the feature matrix (as in node-feature MLPs). Then it combines these hidden representations

through a linear transform $\mathbf{W}$ of their concatenation, with skip connections that add back in $\mathbf{h_A}$ and $\mathbf{h_X}$ to better preserve pure adjacency or node feature information. Finally, it puts this combined representation through a non-linearity and another MLP to make a prediction.

**Separating then mixing adjacency and feature information.** LINKX separately embeds the adjacency $\mathbf{A}$ to $\mathbf{h_A}$ and the features $\mathbf{X}$ into $\mathbf{h_X}$ before mixing them for a few reasons. First, we note that this design is reminiscent of fusion architectures in multimodal networks, where data from different modalities are processed and combined in a neural network [24, 78]. In our setting, we can view adjacency information and node feature information as separate modalities. Since node feature MLPs and LINK do well independently on different datasets, this allows us to preserve their individual performance if needed. Ignoring $\mathbf{h_X}$ information is similar to just using LINK, and ignoring $\mathbf{h_A}$ information is just using an node feature MLP. Still, to preserve the ability to just learn a similar mapping to LINK or to a node feature MLP, we find that having the additive skip connections helps to get performance at least as good as either baseline. Our initial empirical results showed that simply concatenating adjacency and node features as input to a network does worse overall empirically (see Appendix C.1).

There are also computational benefits to our design choices. Embedding $\mathbf{A}$ is beneficial for depth as adding more layers to the MLPs only gives an $\mathcal{O}(d^2)$ cost — depending only on the hidden dimension $d$ — and thus does not scale in the number of edges $|E|$ as when adding layers to message-passing GNNs. This is because the graph information in $\mathbf{A}$ is already compressed to hidden feature vectors after the first linear mapping of $\mathrm{MLP_A}$, and we do not need to propagate along the graph in later steps. Moreover, this enables a sparse-dense matrix product to compute the first linear mapping of $\mathrm{MLP_A}$ on $\mathbf{A}$, which greatly increases efficiency as $\mathbf{A}$ is typically very sparse for real-world graphs. Separate embeddings are key here, as this would not be possible if we for instance concatenated $\mathbf{A}$ and $\mathbf{X}$ when $\mathbf{X}$ is large and dense.

**Simple minibatching.** Message-passing GNNs must take graph topology into account when minibatching with techniques such as neighbor sampling, subgraph sampling, or graph partitioning. However, LINKX does not require this, as it utilizes graph information solely through defining adjacency matrix columns as features. Thus, we can train LINKX with standard stochastic gradient descent variants by taking i.i.d. samples of nodes along with the corresponding columns of the adjacency and feature matrix as features. This is much simpler than the graph minibatching procedures for message-passing GNNs, which require specific hyperparameter choices, have to avoid exponential blowup of number of neighbors per layer, and are generally more complex to implement [77]. In Section 5.3, we use the simple LINKX minibatching procedure for large-scale experiments that show that LINKX with this minibatching style outperforms GNNs with graph minibatching methods. This is especially important on the scale of the wiki dataset, where none of our tested methods — other than MLP — is capable of running on a Titan RTX GPU with 24 GB GPU RAM (see Section 5).

## 4.3 Complexity Analysis

Using the above notation, a forward pass of LINKX has a time complexity of $\mathcal{O}\left(d|E| + nd^2L\right)$, in which $d$ is the hidden dimension (which we assume to be on the same order as the input feature dimension $D$), $L$ is the number of layers, $n$ is the number of nodes, and $|E|$ is the number of edges. We require a $\mathcal{O}(d|E|)$ cost for the first linear mapping of $\mathbf{A}$ and a $\mathcal{O}(d^2)$ cost per layer for MLP operations on hidden features, for $L$ total layers and each of $n$ nodes.

As mentioned above, message passing GNNs have to propagate using the adjacency in each layer, so they have an $L|E|$ term in the complexity. For instance, an $L$-layer GCN [38] with $d$ hidden dimensions has $\mathcal{O}(dL|E| + nd^2L)$ complexity, as it costs $\mathcal{O}(d|E|)$ to propagate features in each layer, and $\mathcal{O}(nd^2)$ to multiply by the weight matrix in each layer.

Non-homophilous methods often make modifications to standard architectures that increase computational cost, such as using higher-order neighborhoods or using additional hidden embeddings [82]. For instance, the complexity of MixHop [1] is $\mathcal{O}(K(dL|E| + nd^2L))$, which has an extra factor $K$ that is the number of adjacency powers to propagate with. The complexity of GCNII [15] is asymptotically the same as that of GCN, but in practice it requires more computations per layer due to residual connections and linear combinations, and it also often achieves best performance with a large number of layers $L$. $\mathrm{H_2GCN}$ [82] is significantly more expensive due to its usage of strict

two-hop neighborhoods, which requires it to form the squared adjacency $\mathbf{A}^2$. This makes the memory requirements intractable even for medium sized graphs (see Section 5).

## 5 Experiments

We conduct two sets of experiments for node classification on our proposed non-homophilous datasets. One set of experiments does full batch gradient descent training for all applicable methods. This of course limits the size of each model, as the large datasets require substantial GPU memory to train on. Our other set of experiments uses minibatching methods. As all graph-based methods run out of memory on the wiki dataset, even on 24 GB GPUs, we only include wiki results in the minibatching section. In all settings, our LINKX model matches or outperforms other methods.

Table 3: Experimental results. Test accuracy is displayed for most datasets, while genius displays test ROC AUC. Standard deviations are over 5 train/val/test splits. The three best results per dataset are highlighted. (M) denotes some (or all) hyperparameter settings run out of memory.

| | Penn94 | pokec | arXiv-year | snap-patents | genius | twitch-gamers |
|---|---|---|---|---|---|---|
| MLP | $73.61 \pm 0.40$ | $62.37 \pm 0.02$ | $36.70 \pm 0.21$ | $31.34 \pm 0.05$ | $86.68 \pm 0.09$ | $60.92 \pm 0.07$ |
| L Prop 1-hop | $63.21 \pm 0.39$ | $53.09 \pm 0.05$ | $43.42 \pm 0.17$ | $30.28 \pm 0.09$ | $66.02 \pm 0.16$ | $62.77 \pm 0.24$ |
| L Prop 2-hop | $74.13 \pm 0.46$ | $76.76 \pm 0.03$ | $46.07 \pm 0.15$ | $38.61 \pm 0.07$ | $67.04 \pm 0.20$ | $63.88 \pm 0.24$ |
| LINK | $80.79 \pm 0.49$ | $80.54 \pm 0.03$ | $53.97 \pm 0.18$ | $60.39 \pm 0.07$ | $73.56 \pm 0.14$ | $64.85 \pm 0.21$ |
| SGC 1-hop | $66.79 \pm 0.27$ | $53.61 \pm 0.17$ | $32.83 \pm 0.13$ | $30.31 \pm 0.06$ | $82.36 \pm 0.37$ | $58.97 \pm 0.19$ |
| SGC 2-hop | $76.09 \pm 0.45$ | $62.81 \pm 1.42$ | $32.27 \pm 0.06$ | $29.09 \pm 0.09$ | $82.10 \pm 0.14$ | $59.94 \pm 0.21$ |
| C&S 1-hop | $74.28 \pm 1.19$ | $62.35 \pm 0.06$ | $44.51 \pm 0.16$ | $35.55 \pm 0.05$ | $82.93 \pm 0.15$ | $64.86 \pm 0.27$ |
| C&S 2-hop | $78.40 \pm 3.12$ | $81.69 \pm 0.09$ | $49.78 \pm 0.26$ | $49.08 \pm 0.04$ | $84.94 \pm 0.49$ | $65.02 \pm 0.16$ |
| GCN | $82.47 \pm 0.27$ | $75.45 \pm 0.17$ | $46.02 \pm 0.26$ | $45.65 \pm 0.04$ | $87.42 \pm 0.37$ | $62.18 \pm 0.26$ |
| GAT | $81.53 \pm 0.55$ | $71.77 \pm 6.18$ (M) | $46.05 \pm 0.51$ | $45.37 \pm 0.44$ (M) | $55.80 \pm 0.87$ | $59.89 \pm 4.12$ |
| GCNJK | $81.63 \pm 0.54$ | $77.00 \pm 0.14$ | $46.28 \pm 0.29$ | $46.88 \pm 0.13$ | $89.30 \pm 0.19$ | $63.45 \pm 0.22$ |
| GATJK | $80.69 \pm 0.36$ | $71.19 \pm 6.96$ (M) | $45.80 \pm 0.72$ | $44.78 \pm 0.50$ | $56.70 \pm 2.07$ | $59.98 \pm 2.87$ |
| APPNP | $74.33 \pm 0.38$ | $62.58 \pm 0.08$ | $38.15 \pm 0.26$ | $32.19 \pm 0.07$ | $85.36 \pm 0.62$ | $60.97 \pm 0.10$ |
| H$_2$GCN | (M) | (M) | $49.09 \pm 0.10$ | (M) | (M) | (M) |
| MixHop | $83.47 \pm 0.71$ | $81.07 \pm 0.16$ | $51.81 \pm 0.17$ | $52.16 \pm 0.09$ (M) | $90.58 \pm 0.16$ | $65.64 \pm 0.27$ |
| GPR-GNN | $81.38 \pm 0.16$ | $78.83 \pm 0.05$ | $45.07 \pm 0.21$ | $40.19 \pm 0.03$ | $90.05 \pm 0.31$ | $61.89 \pm 0.29$ |
| GCNII | $82.92 \pm 0.59$ | $78.94 \pm 0.11$ (M) | $47.21 \pm 0.28$ | $37.88 \pm 0.69$ (M) | $90.24 \pm 0.09$ | $63.39 \pm 0.61$ |
| LINKX | $84.71 \pm 0.52$ | $82.04 \pm 0.07$ | $56.00 \pm 1.34$ | $61.95 \pm 0.12$ | $90.77 \pm 0.27$ | $66.06 \pm 0.19$ |

### 5.1 Experimental Setup

**Methods.** We include both methods that are graph-agnostic and node-feature-agnostic as simple baselines. The node-feature-agnostic models of two-hop label propagation [57] and LINK (logistic regression on the adjacency matrix) [79] have been found to perform well in various non-homophilous settings, but they have often been overlooked by recent graph representation learning work. Also, we include SGC [71] and C&S [32] as simple, scalable methods that perform well on homophilic datasets. We include a two-hop propagation variant of C&S in analogy with two-step label propagation. In addition to representative general GNNs, we also include GNNs recently proposed for non-homophilous settings. The full list of methods is: **Only node features:** MLP [26]. **Only graph topology:** label propagation (standard and two-hop) [80, 57], LINK [79]. **Simple methods:** SGC [71], C&S [32] and their two-hop variants. **General GNNs:** GCN [38], GAT [69], jumping knowledge networks (GCNJK, GATJK) [72], and APPNP [39]. **Non-homophilous methods:** H$_2$GCN [82], MixHop [1], GPR-GNN [17], GCNII [15], and LINKX (ours).

**Minibatching methods.** We also evaluate GNNs with various minibatching methods. We take GC-NJK [72] and MixHop [1] as our base models for evaluation, as they are representative of many GNN design choices and MixHop performs very well in full batch training. As other minibatching methods are trickier to make work with these models, we use the Cluster-GCN [16] and GraphSAINT [77] minibatching methods, which sample subgraphs. We include both the node based sampling and random walk based sampling variants of GraphSAINT. We compare these GNNs with MLP, LINK, and our LINKX, which use simple i.i.d. node minibatching.

**Training and evaluation.** Following other works in non-homophilous graph learning evaluation, we take a high proportion of training nodes [82, 58, 73]; we run each method on the same five random 50/25/25 train/val/test splits for each dataset. All methods requiring gradient-based optimization are run for 500 epochs, with test performance reported for the learned parameters of highest validation performance. We use ROC-AUC as the metric for the class-imbalanced genius dataset (about 80% of nodes are in the majority class), as it is less sensitive to class-imbalance than accuracy. For other datasets, we use classification accuracy as the metric. Further experimental details are in Appendix B.

## 5.2  Full-Batch Results

Table 3 lists the results of each method across the datasets that we propose. Our datasets reveal several important properties of non-homophilous node classification. Firstly, the stability of performance across runs is better for our datasets than those of Pei et al. [58] (see [82] results). Secondly, as suggested by prior theory and experiments [82, 1, 17], the non-homophilous GNNs usually do well — though not necessarily on every dataset.

The core assumption of homophily in SGC and C&S that enables them to be simple and efficient does not hold on these non-homophilous datasets, and thus the performance of these methods is typically relatively low. Still, as expected, two-hop variants generally improve upon their one-hop counter-parts in these low-homophily settings.

One consequence of using larger datasets for benchmarks is that the tradeoff between scalability and learning performance of non-homophilous methods has become starker, with some methods facing memory issues. This tradeoff is especially important to consider in light of the fact that many scalable graph learning methods rely on implicit or explicit homophily assumptions [71, 32, 20, 10], and thus face issues when used in non-homophilous settings.

Finally, LINKX achieves superior performance on all datasets, taking advantage of LINK's power, while also being able to utilize node features where they provide additional information.

## 5.3  Minibatching Results

Table 4: Minibatching results on our proposed datasets. † denotes that 10 random partitions of the graphs are used for testing GraphSAINT sampling. (T) denotes that five runs takes $\geq 48$ hours for a single hyperparameter setting. Best results up to a standard deviation are highlighted.

| | Penn94 | pokec † | arXiv-year | snap-patents † | genius | twitch-gamers † | wiki † |
|---|---|---|---|---|---|---|---|
| MLP Minibatch | 74.24±0.55 | 62.14±0.05 | 36.89±0.11 | 22.96±0.81 | 82.35±0.38 | 61.01±0.06 | 37.38±0.21 |
| LINK Minibatch | 81.61±0.34 | 81.15±0.25 | 53.76±0.28 | 45.65±8.25 | 80.95±0.07 | 64.38±0.26 | 57.11±0.26 |
| GCNJK-Cluster | 69.99±0.85 | 72.67±0.05 | 44.05±0.11 | 37.62±0.31 | 83.04±0.56 | 61.15±0.16 | (T) |
| GCNJK-SAINT-Node | 72.80±0.43 | 63.68±0.06 | 44.30±0.22 | 26.97±0.10 | 80.96±0.09 | 59.50±0.35 | 44.86±0.19 |
| GCNJK-SAINT-RW | 72.29±0.49 | 65.00±0.11 | 47.40±0.17 | 33.05±0.06 | 81.04±0.14 | 59.82±0.27 | 47.39±0.19 |
| MixHop-Cluster | 75.79±0.44 | 76.67±0.07 | 48.41±0.31 | 46.82±0.11 | 81.12±0.10 | 62.95±0.08 | (T) |
| MixHop-SAINT-Node | 75.61±0.55 | 66.42±0.06 | 44.84±0.18 | 27.45±0.11 | 81.06±0.08 | 59.58±0.27 | 47.39±0.18 |
| MixHop-SAINT-RW | 76.38±0.50 | 67.92±0.06 | 50.55±0.20 | 34.21±0.07 | 82.25±0.78 | 60.39±0.16 | 49.15±0.26 |
| LINKX Minibatch | 84.50±0.65 | 81.27±0.38 | 53.74±0.27 | 60.27±0.29 | 85.81±0.10 | 65.84±0.19 | 59.80±0.41 |

Our experimental results for minibatched methods on our proposed datasets are in Table 4. Since GraphSAINT does not partition the nodes of the graph into subgraphs that cover all nodes, we test on the full input graph for the smaller datasets and uniformly random partitions of the graph into 10 induced subgraphs for the larger datasets.

First, we note that both Cluster-GCN and GraphSAINT sampling lead to performance degradation for these methods on our proposed non-homophilous datasets. When compared to the full-batch training results of the previous section, classification accuracy is typically substantially lower. Further experiments in Appendix C.2 give evidence that the performance degradation is often more substantial in non-homophilous settings, and provides possible explanations for why this may be the case.

On the other hand, LINKX does not suffer much performance degradation with the simple i.i.d. node minibatching technique. In fact, it matches or outperforms all methods in this setting, often by a wide margin. Though LINK performs on par with LINKX in arXiv-year and pokec, our LINKX model significantly outperforms it on other datasets, again due to LINKX's ability to integrate node feature information. We again stress that the LINKX minibatching is very simple to implement, yet it still substantially outperforms other methods. Consequently, LINKX is generally well-suited for scalable node classification across a broad range of non-homophilous settings, surpassing even specially designed non-homophilous GNNs with current graph minibatching techniques.

## 6  Discussion and Conclusion

In this paper, we propose new, high-quality non-homophilous graph learning datasets, and we benchmark simple baselines and representative graph representation learning methods across these datasets. Further, we develop LINKX: a strong, simple, and scalable method for non-homophilous

classification. Our experiments show that LINKX significantly outperforms other methods on our proposed datasets, thus providing one powerful method in the underexplored area of scalable learning on non-homophilous graphs. We hope that our contributions will provide researchers with new avenues of research in learning on non-homophilous graphs, along with better tools to test models and evaluate utility of new techniques.

While we do find utility in our proposed datasets and LINKX model, this work is somewhat limited by only focusing on transductive node classification. This setting is the most natural for studying performance in the absence of homophily, since here we define homophily in terms of the node labels, and previous non-homophilous GNN work using the Pei et al. [58] data also studies this setting exclusively [82, 17]. Using other Facebook 100 datasets besides Penn94 [67] would allow for inductive node classification, but LINKX does not directly generalize to this setting. Our proposed datasets and model LINKX could be used for link prediction, but this is left for future work.

**Broader Impact.** Fundamental research in graph learning on non-homophilous graphs has the potential for positive societal benefit. As a major application, it enables malicious node detection techniques in social networks and transaction networks that are not fooled by fraudsters' connections to legitimate users and customers. This is a widely studied task, and past works have noted that non-homophilous structures are present in many such networks [11, 25, 55]. We hope that this paper provides insight on the homophily limitations of existing scalable graph learning models and help researchers design scalable models that continue to work well in the non-homophilous regime, thus improving the quality of node classification on graphs more broadly. As our proposed datasets have diverse structures and our model performs well across all of these datasets, the potential for future application of our work to important non-homophilous tasks is high.

Nevertheless, our work could also have potential for different types of negative social consequences. Nefarious behavior by key actors could be one source of such consequences. Nonetheless, we expect that the actors that can make use of large-scale social networks for gender prediction as studied in our work are limited in number. Actors with both the capability and incentive to perform such operations probably mostly consist of entities with access to large social network data such as social media companies or government actors with auxiliary networks [50]. Smaller actors can perform certain attacks, but this may be made more difficult by resource requirements such as the need for certain external information [50] or the ability to add nodes and edges before an anonymized version of a social network is released [5]. Furthermore, additional actors could make use of deanonymization attacks [30, 49, 50] to reveal user identities in supposedly anonymized datasets.

Also, accidental consequences and implicit biases are a potential issue, even if the applications of the learning algorithms are benign and intended to benefit society [47]. Performance of algorithms may vary substantially between intersectional subgroups of subjects — as in the case of vision-based gender predictors [12] (and some have questioned the propriety of vision-based gender classifiers altogether). Thus, there may be disparate effects on different populations, so care should be taken to understand the impact of those differences across subgroups. Moreover, large datasets require computing resources, so projects can only be pursued by large entities at the possible expense of the individual and smaller research groups [8]. This is alleviated by the fact that our experiments are each run on one GPU, and hence have significantly less GPU computing requirements than much current deep learning research. Thus, smaller research groups and independent researchers should find our work beneficial, and should be able to build on it.

Finally, the nature of collection of online user information also comes with notable ethical concerns. Common notice-and-consent policies are often ineffective in actually protecting user privacy [52]. Indeed, users may not actually have much choice in using certain platforms or sharing data due to social or economic reasons. Also, users are generally unable to fully read and understand all of the different privacy policies that they come across, and may not understand the implications of having their data available for long periods of time to entities with powerful inference algorithms. Furthermore, people may rely on obscurity for privacy [29], but this assumption may be ignored in courts of law, and it may be directly broken when data leaks or is released in aggregated form without sufficient privacy protections. Overall, while we believe that our work will benefit machine learning research and enable positive applications, we must still be aware of possible negative consequences.

**Acknowledgements**

We thank Abhay Singh, Austin Benson, and Horace He for insightful discussions. We also thank the rest of Cornell University Artificial Intelligence for their support and discussion. We thank Facebook AI for funding equipment that made this work possible.

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
