

(a) $h = 1,\ \hat{h} = 1$    (b) $h = 0,\ \hat{h} = 0$    (c) $h = .5,\ \hat{h} = 0$    (d) $h = .33,\ \hat{h} = 0$

(e) $h = .53,\ \hat{h} = .07$    (f) $h = .66,\ \hat{h} = .04$

Figure 2: Examples of graphs with different label-topology relationships and comparison of our measure $\hat{h}$ with the edge homophily ratio $h$. The node classes are labeled by color. Pink edges link nodes of the same class, while purple edges link nodes of different classes. (a,b) Pure homophily and pure heterophily. Both measures equal 1 in homophily and 0 in heterophily. (c,d) Graphs where each node is connected to one member of each class. Edge homophily depends on the number of classes, while our measure $\hat{h}$ does not. (e,f) Random Erdős-Rényi graphs in which edges are independent of labels. Edge homophily is sensitive to class imbalance, while our measure $\hat{h}$ is not.

## A    Compatibility Matrices and Statistics

### A.1    Measuring Homophily

In Section 3.2, we consider metrics that attempt to capture the level of class-label homophily in a graph in a single scalar quantity. These metrics could be useful for practitioners who need to choose appropriate graph learning algorithms for some graph data that they have — performance of algorithms heavily depends on the homophily of the graph. Moreover, they are useful for choosing datasets to benchmark on, as we do in this paper.

A better representation of homophily may be given by the compatibility matrix, which consists of $C^2$ values instead of a single scalar value. Following previous work [82], for a graph $G$ with $C$ node classes we define the $C \times C$ compatibility matrix $\mathbf{H}$ by

$$\mathbf{H}_{kl} = \frac{|(u, v) \in E : k_u = k,\ k_v = l|}{|(u, v) \in E : k_u = k|}. \tag{9}$$

This captures finer details of label-topology relationships in graphs than single scalar metrics capture. For classes $k$ and $l$, the entry $\mathbf{H}_{kl}$ measures the proportion of edges from nodes of class $k$ that are connected to nodes of class $l$. A homophilous graph has high values of $\mathbf{H}_{kk}$ for each class $k$.

Compatibility matrices for our proposed datasets are shown in Figure 3. As evidenced by the different patterns, the proposed datasets show interesting types of label-topology relationships besides homophily. For instance, the citation datasets arXiv-year and snap-patents have primarily lower-triangular structure, since most citations reference past work. In wiki, low view articles tend to link to each other, while mid-rank articles frequently link to highly viewed articles and vice versa. In Pokec, there is some heterophily, in that one gender has some preference for friends of another gender.

However, the compatibility matrix can be unwieldy for datasets with many classes. For the cases where a single scalar representation of homophily is desired, our introduced measure better captures the presence or absence of homophily than existing metrics. Figure 2 compares our measure $\hat{h}$ to edge homophily on example graphs.

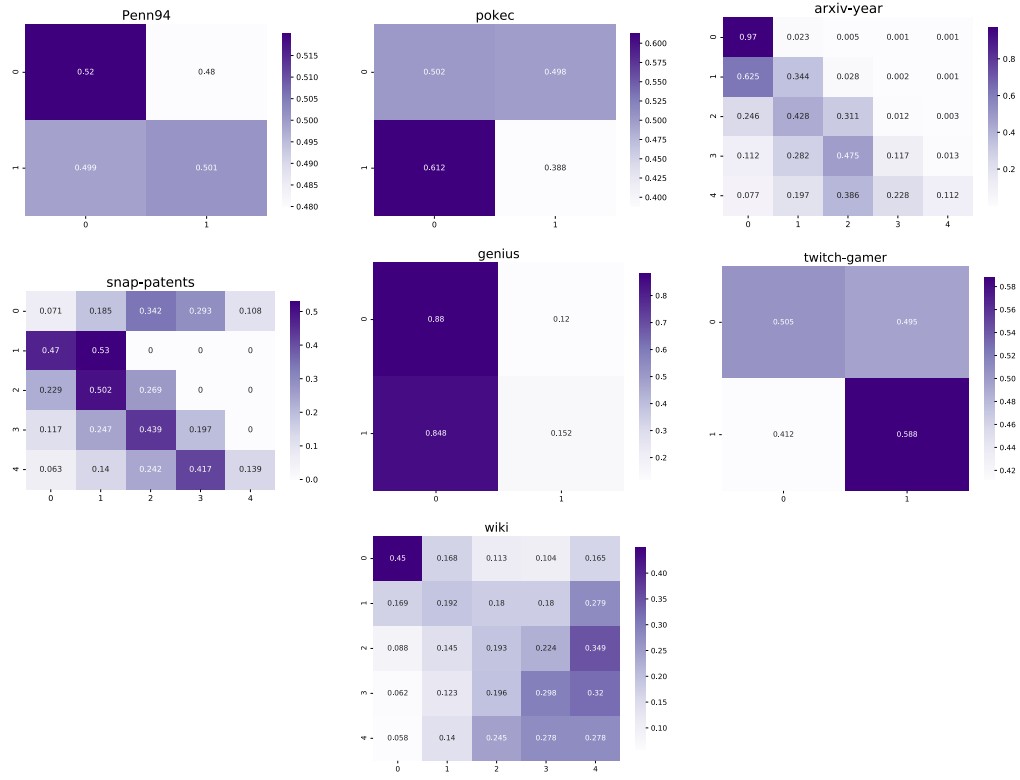

Figure 3: Compatibility matrices of our proposed datasets. These datasets from a variety of different contexts exhibit a wide range of non-homophilous structures.

## A.2 Homophilous Data Statistics

Table 5: Statistics for homophilic graph datasets. # C is the number of node classes.

| Dataset | # Nodes | # Edges | # C | Edge hom. | $\hat{h}$ (ours) |
|---|---|---|---|---|---|
| Cora | 2,708 | 5,278 | 7 | .81 | .766 |
| Citeseer | 3,327 | 4,552 | 6 | .74 | .627 |
| Pubmed | 19,717 | 44,324 | 3 | .80 | .664 |
| ogbn-arXiv | 169,343 | 1,166,243 | 40 | .66 | .416 |
| ogbn-products | 2,449,029 | 61,859,140 | 47 | .81 | .459 |
| oeis | 226,282 | 761,687 | 5 | .50 | .532 |

In contrast to the different compatibility matrix structures of our proposed non-homophilous datasets, much other graph data have primarily homophilous relationships, as can be seen in Figure 4 and Table 5. The Cora, CiteSeer, PubMed, ogbn-arXiv, and ogbn-products datasets are widely used as benchmarks for node classification [74, 31], and are highly homophilous, as can be seen by the diagonally dominant structure of the compatibility matrices and by the high edge homophily and $\hat{h}$.

We collected the oeis dataset displayed in the bottom right of Figure 4. The nodes are entries in the Online Encyclopedia of Integer Sequences [64], and directed edges link an entry to any other entry that it cites. In analogy to arXiv-year and snap-patents, the node labels are the time of posting of the sequence. However, in this case the graph relationships are homophilous, even as we vary the number of distinct classes (time periods). This is in part due to differences between posting in this online encyclopedia and publication of academic papers or patents. For instance, there is less overhead to posting an entry in the OEIS, so users often post separate related entries and variants of these entries in rapid succession. Also, an entry in the encyclopedia often inspires other people to work on similar entries, which can be created in much less time than an academic follow-up work to a given paper.

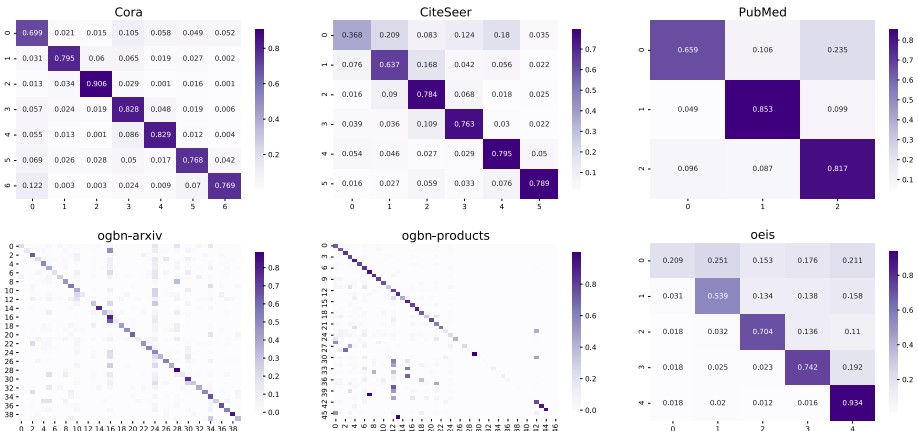

Figure 4: Compatibility matrices of homophilic datasets. The diagonal dominance indicates strong homophily.

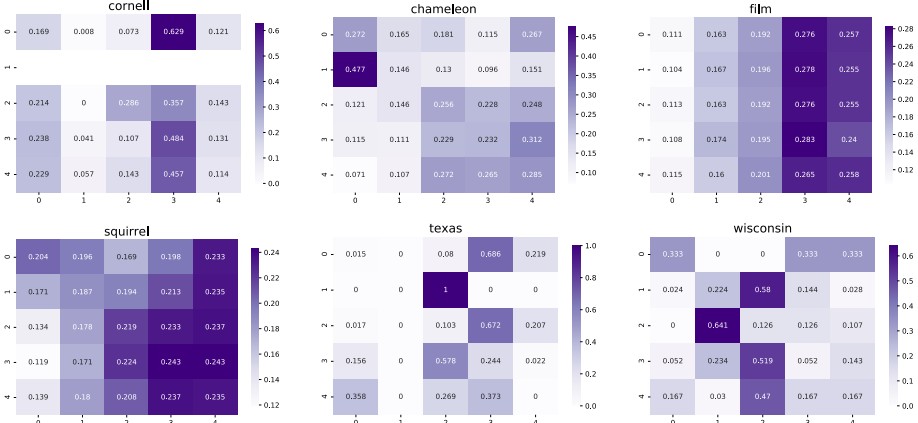

Figure 5: Compatibility matrices of datasets in Pei et al. [58] (collected by [61, 66, 48]). The "film" dataset is also referred to as "Actor". Note that there are no edges leading out of the nodes of class 1 in the Cornell dataset, so there is an empty row in its matrix.

These related entries tend to cite each other, which contributes to homophilic relationships over time. Thus, the data here does not follow the special temporal citation structure of academic publications and patents.

### A.3   Previous Non-Homophilous Data

For the six datasets in Pei et al. [58] often used in evaluation of graph representation learning methods in non-homophilous regimes [82], basic statistics are listed in Table 1 and compatibility matrices are displayed in Figure 5. We propose datasets that have up to orders of magnitude more nodes and edges and come from a wider range of contexts. There are several cases of class-imbalance in these previously used datasets, which may make the edge homophily misleading. As discussed in Appendix A.1, our measure may be able to alleviate issues with edge homophily in measuring homophily of these datasets, and offers a way to distinguish between the Chameleon, Actor, and Squirrel datasets that all have similar edge homophily.

### A.4   General Non-homophilous Settings

Different settings in which non-homophilous / disassortative relationships are prevalent have been identified in the literature, and many of these non-homophilous settings are represented by our

proposed datasets. We list these general non-homophilous settings for reference, with our proposed datasets that belong to these settings in parentheses:

- Gender relations in social or interaction networks [3, 18, 34] (Penn94, Pokec).
- Technological and internet relationships, such as in web page connections [51, 58] (wiki).
- Malicious or fraudulent nodes, such as in auction networks [13, 55] (genius).
- Publication time in citation networks [57] (arXiv-year, snap-patents).
- Biological structures such as in food webs [25] and protein interactions [51].
- Specific online user attributes [60] (twitch gamers)

While not all example graph data from these contexts are non-homophilous, a diverse range are. In order to succeed in future applications in these contexts, it is of importance to develop methods that are able to handle non-homophilous structures.

## A.5    Class-Imbalance and Metrics

In this section, we present experiments that demonstrate an instance in which our metric is not affected by imbalanced classes, while edge homophily is. We generate graphs in which node labels are independent of edges by randomly choosing node labels and generating graph edges by the Erdős-Rényi random graph model [22]. In particular, we fix the number of classes to two, the number of nodes to 100, and the probability of edge formation as .25 between every pair of nodes. Then we generate 100 samples of these random graphs, and compute the mean and standard deviation of both edge homophily $h$ and our measure $\hat{h}$. As seen in Figure 6, our measure $\hat{h}$ is constantly near zero as we increase the size of the majority class, while the edge homophily $h$ increases as the size of majority class increases.

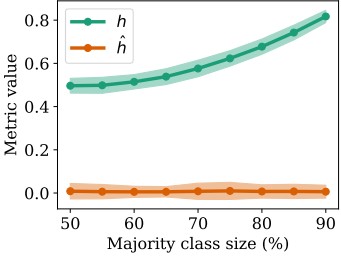

Figure 6: Comparison of edge homophily $h$ and our measure $\hat{h}$ on random class-imbalanced graph data with edges independent of node labels. Three standard deviations are shaded. Our measure is mostly constant as the classes become more imbalanced, while edge homophily increases.

## A.6    Degree Distributions of Proposed Non-homophilous datasets

Past work has found that the degree distribution can affect the performance of models on node classification [82]. As a result, we provide degree distributions of all of our proposed non-homophilous datasets in Figure 7. The degree distributions are all heavy-tailed, as is typically expected in real-world graph data. The wiki distribution also has an interesting additional property, where the mode appears to be at a degree between 10 and 100.

## A.7    Two-hop homophily levels

While the homophily measures are based on one-hop information, we may also consider properties of two-hop neighborhoods. Even though there are limitations to measuring higher-order homophily [68], here we estimate an interesting quantity in two-hop neighborhoods. We use a node homophily measure, where the neighborhood of each node is defined to be the nodes of exactly two hops away:

$$\frac{1}{|V|} \sum_{v \in V} \frac{\text{Number of exact two-hop neighbors of } v \text{ with same class as } v}{\text{Number of } v\text{'s exact two-hop neighbors}}. \tag{10}$$

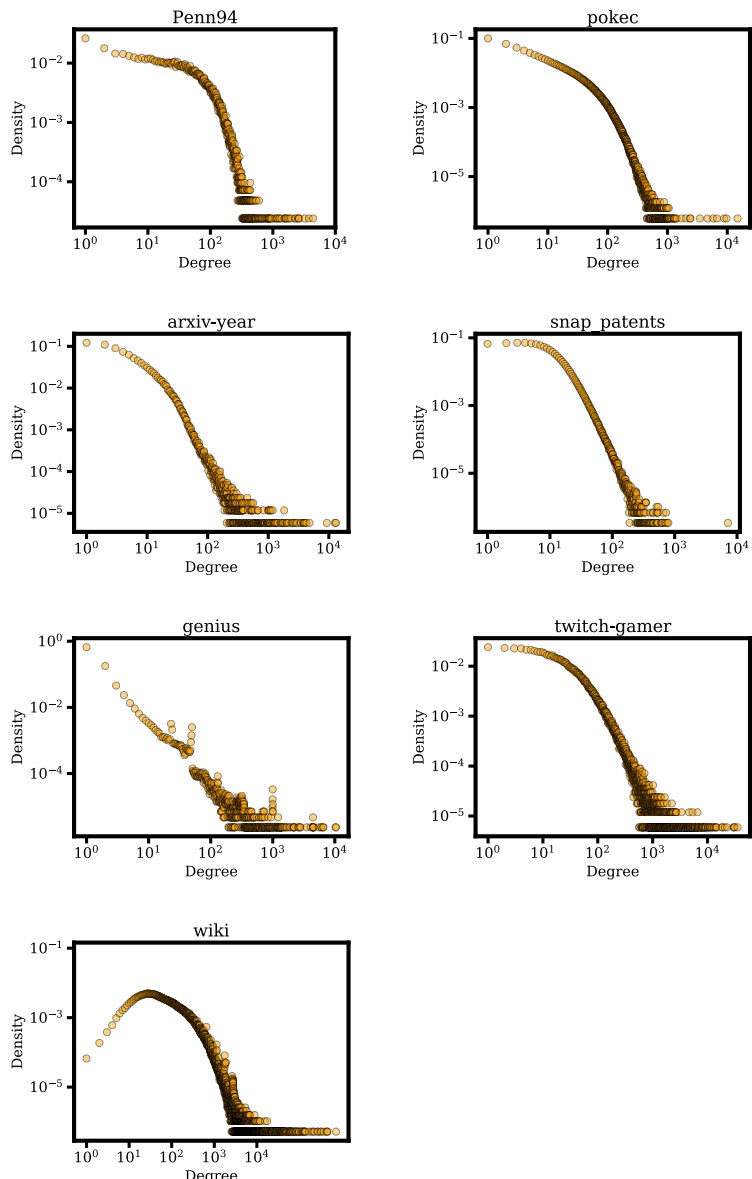

Figure 7: Degree distribution of our proposed non-homophilous datasets

This is an expensive operation to compute, so we approximate the sum over a random sample of $k$ nodes, and then replace the normalization factor of $\frac{1}{|V|}$ by $\frac{1}{k}$. Note that for each node $v$ in the sample, the inner sum is still using the two-hop neighborhoods of the original graph. Table 6 compares the computed one-hop and two-hop measures. The two-hop measure is mostly similar to or somewhat higher than the standard one-hop node homophily.

Table 6: Comparison of node homophily over one-hop neighborhoods and strict two-hop neighborhoods. The measure over two-hop neighborhoods is estimated with $k = 500$ sampled nodes.

| Dataset | Two-hop | One-hop |
|---|---|---|
| Penn94 | .474 | .483 |
| pokec | .611 | .428 |
| arXiv-year | .341 | .289 |
| snap-patents | .330 | .221 |
| genius | .749 | .508 |
| twitch-gamers | .513 | .556 |

## B   Experimental Details

For gradient-based optimization, we use the AdamW optimizer [37, 45] with weight decay .001 and learning rate .01 by default, unless we tune the optimizer for a particular method (as noted below in B.1). Hyperparameter tuning is conducted using grid search for most methods. Tuning for C&S is done as in the original paper [32], which uses Optuna [2] for Bayesian hyperparameter optimization. All graphs are treated as undirected besides arXiv-year and snap-patents, in which the directed nature of the edges capture useful temporal information; however, we find that label propagation and C&S (which builds on label propagation) perform better with undirected graphs in these cases, so we keep the graphs as undirected for these methods.

We implement our methods and run experiments in PyTorch [56] (3-clause BSD license), and make heavy use of the PyTorch-Geometric library [23] (MIT license) for graph representation learning. For full-batch training, simple methods are run on a NVIDIA 2080 Ti with 11 GB GPU memory. In cases where the NVIDIA 2080Ti did not provide enough memory, we re-ran experiments on a NVIDIA Titan RTX with 24 GB GPU memory, reporting (M) in Table 3 if the GPU memory was still insufficient. For minibatch training on wiki, we also make use of NVIDIA RTX 3090 GPUs with 24 GB GPU memory. Each experiment was run on one GPU at a time.

### B.1   Full-batch Hyperparameters

Experimental results for full-batch training are reported on the hyperparameter settings below, where we choose the settings that achieve the highest performance on the validation set. We choose hyperparameter grids that do not necessarily give optimal performance, but hopefully cover enough regimes so that each model is reasonably evaluated on each dataset. Unless otherwise stated, each GNN has dropout of .5 [65] and BatchNorm [33] in each layer. The hyperparameter grids for the different methods are:

- MLP: hidden dimension $\in \{16, 32, 64, 128, 256\}$, number of layers $\in \{2, 3\}$. We use ReLU activations.
- Label propagation: $\alpha \in \{.01, .1, .25, .5, .75, .9, .99\}$. We use 50 propagation iterations.
- LINK: weight decay $\in \{.001, .01, .1\}$.
- SGC: weight decay $\in \{.001, .01, .1\}$.
- C&S: Normalized adjacency matrix $A_1, A_2 \in \{D^{-\frac{1}{2}}AD^{-\frac{1}{2}}, D^{-1}A, AD^{-1}\}$ for the residual propagation and label propagation, where $A$ is the adjacency matrix of the graph and $D$ is the diagonal degree matrix; $\alpha_1, \alpha_2 \in (0.0, 1.0)$ for the two propagations. Both Autoscale and FDiff-scale were used for all experiments, and scale $\in (0.1, 10.0)$ was searched in FDiff-scale settings. The base predictor is chosen as the best MLP model for each dataset.

- GCN: lr $\in \{.1, .01, .001\}$, hidden dimension $\in \{4, 8, 16, 32, 64\}$, except for snap-patents and pokec, where we omit hidden dimension = 64. Each activation is a ReLU. 2 layers were used for all experiments.

- GAT: lr $\in \{.1, .01, .001\}$. For snap-patents and pokec: hidden channels $\in \{4, 8, 12\}$ and gat heads $\in \{2, 4\}$. For all other datasets: hidden channels $\in \{4, 8, 12, 32\}$ and gat heads $\in \{2, 4, 8\}$. We use the ELU activation [19]. 2 layers were used for all experiments.

- GCNJK: Identical for GCN, also including JK Type $\in \{$cat, max$\}$.

- GATJK: Identical for GAT, also including JK Type $\in \{$cat, max$\}$.

- APPNP: MLP hidden dimension $\in \{16, 32, 64, 128, 256\}$, learning rate $\in \{.01, .05, .002\}$, $\alpha \in \{.1, .2, .5, .9\}$. We truncate the series at the $K = 10$th power of the adjacency.

- H$_2$GCN: hidden dimension $\in \{8, 16, 32, 64\}$, number of layers $\in \{1, 2\}$, dropout $\in \{0, .5\}$. The architecture follows Section 3.2 of [82].

- MixHop: hidden dimension $\in \{8, 16, 32\}$, number of layers $\in \{2, 3\}$. Each layer has uses the 0th, 1st, and 2nd powers of the adjacency and has ReLU activations. The last layer is a linear projection layer, instead of the attention output mechanism in [1].

- GPR-GNN: The basic setup and grid is the same as that of APPNP. We use their Personalized PageRank weight initialization.

- GCNII: number of layers $\in \{2, 8, 16, 32, 64\}$, strength of the initial residual connection $\alpha_\ell \in \{0.1, 0.2, 0.5\}$, hyperparameter used to compute the strength of the identity mapping $\lambda \in \{0.5, 1.0, 1.5\}$.

- LINKX: We use take MLP$_\mathbf{A}$ and MLP$_\mathbf{X}$ to be one layer networks, i.e. linear mappings of size $d \times n$ and $d \times D$, respectively. The hidden dimension is taken to be $d \in \{16, 32, 128, 256\}$, and the number of layers of MLP$_f$ is in $\{1, 2, 3\}$.

## B.2    Minibatching hyperparameters

The setup for minibatching is similar to the setup of full-batch training as above, with some differences that we note here. For all GNNs, we fix the hidden dimension to 128, which is a common hidden dimension used in Cluster-GCN [16] and GraphSAINT [77]. We use concatenation jumping knowledge connections [72] for GCNJK. For GCNJK and MixHop, our hyperparameter grid only chooses a number of layers $L \in \{2, 4\}$, along with the hyperparameters for the minibatching methods that we give below.

- MLP, LINK, and LINKX use the same hyperparameter grids as in the full-batch setting, and they are trained with standard minibatching with a batch size of $n/10$. We train with one such batch in each of 500 epochs.

- All Cluster-GCN based experiments partition the graph into 200 parts and process a number of parts in $\{1, 5\}$ at once. We train over all partitions in each of 500 epochs.

- All GraphSAINT-Node based experiments have a budget of nodes in $\{5,000, 10,000\}$. We train over five subgraphs for each of 500 epochs.

- All GraphSAINT-RandWalk based experiments use a random walk length of the same size as the number of layers $L$ of the GNN, and use a number of roots in $\{5,000, 10,000\}$. We train over five subgraphs for each of 500 epochs.

These hyperparameter settings are in the same range as those used in the original papers [16, 77] for datasets that have similar node counts to our proposed datasets. For a total number of nodes $n$ in the full graph, note that the MLP, LINK, and LINKX minibatching method processes about $50n$ total nodes across all batches in training. The Cluster-GCN method processes $500n$ nodes. When the number of nodes is 10,000, the GraphSAINT-Node method approximately processes between $601n$ nodes (Penn94) and $8.5n$ nodes (snap-patents). The GraphSAINT-RandWalk method with 10,000 root nodes processes a few times more nodes (usually between two to five times more) than the GraphSAINT-Node method, as more nodes are sampled from the random walks (and we take the number of layers $L$ to be 2 or 4).

# C  Further Experiments

## C.1  Benefits of Separate Embeddings

Table 7: Comparing separately embedding $\mathbf{A}$ and $\mathbf{X}$ (as in LINKX) with direct concatenation. LINKX outperforms the concatenation based model.

|  | Penn94 | pokec | arXiv-year | snap-patents | genius | twitch-gamers |
|---|---|---|---|---|---|---|
| MLP | $73.61 \pm 0.40$ | $62.37 \pm 0.02$ | $36.70 \pm 0.21$ | $31.34 \pm 0.05$ | $86.68 \pm 0.09$ | $60.92 \pm 0.07$ |
| LINK | $80.79 \pm 0.49$ | $80.54 \pm 0.03$ | $53.97 \pm 0.18$ | $60.39 \pm 0.07$ | $73.56 \pm 0.14$ | $64.85 \pm 0.21$ |
| MLP($[\mathbf{A}; \mathbf{X}]$) | $84.64 \pm 0.33$ | $81.74 \pm 0.15$ | $54.15 \pm 0.20$ | $59.12 \pm 0.29$ | $91.61 \pm 0.05$ | $64.89 \pm 0.18$ |
| LINKX | $84.71 \pm 0.52$ | $82.04 \pm 0.07$ | $56.00 \pm 1.34$ | $61.95 \pm 0.12$ | $90.77 \pm 0.27$ | $66.06 \pm 0.19$ |

Here, we give more evidence to justify separately embedding $\mathbf{A}$ and $\mathbf{X}$ in LINKX. Recall from Section 4.2 that there are computational benefits to separately embedding them. Table 7 give results for concatenating $\mathbf{A}$ and $\mathbf{X}$ so that they are jointly embedded as MLP($[\mathbf{A}; \mathbf{X}]$). For this concatenation model, we search over the same hyperparameter grid as that of $\text{MLP}_f$ in LINKX. We see that LINKX generally outperforms the concatenation model (besides on the genius dataset). Moreover, LINKX always outperforms both MLP and LINK, while the concatenation model does not do as well as LINK on snap-patents and is within a standard deviation on twitch-gamers.

## C.2  Minibatching experiments

Table 8: Comparison of different methods on the arXiv graphs with (homophilous) ogbn-arXiv subject labels and (non-homophilous) arXiv-year publication year labels. Percent relative error of minibatch methods against corresponding full batch methods are in parentheses.

|  | ogbn-arXiv Coarse | arXiv-year Coarse |
|---|---|---|
| GCNJK-Full | 70.33 | 44.18 |
| GCNJK-Cluster | 68.28 (2.9%) | 42.88 (3.0%) |
| GCNJK-SAINT-Node | 67.69 (3.7%) | 41.48 (6.1%) |
| GCNJK-SAINT-RW | 69.57 (1.1%) | 43.58 (1.4%) |
| MixHop-Full | 72.26 | 48.14 |
| MixHop-Cluster | 70.16 (3.0%) | 46.90 (2.8%) |
| MixHop-SAINT-Node | 67.26 (7.1%) | 41.68 (14.7%) |
| MixHop-SAINT-RW | 71.54 (1.0%) | 45.94 (5.0%) |

|  | ogbn-arXiv Fine | arXiv-year Fine |
|---|---|---|
| GCNJK-Full | 70.33 | 44.18 |
| GCNJK-Cluster | 68.01 (3.3%) | 42.49 (3.8%) |
| GCNJK-SAINT-Node | 66.18 (5.9%) | 39.88 (9.7%) |
| GCNJK-SAINT-RW | 68.96 (1.9%) | 42.47 (3.9%) |
| MixHop-Full | 72.26 | 48.14 |
| MixHop-Cluster | 69.37 (3.5%) | 46.60 (3.2%) |
| MixHop-SAINT-Node | 64.71 (10.1%) | 39.17 (18.7%) |
| MixHop-SAINT-RW | 70.48 (2.0%) | 43.29 (10.1%) |

**Homophilous vs Non-Homophilous.** To see differences in graph minibatching between homophilous and non-homophilous settings, we setup experiments using the same graph with homophilous labels and then non-homophilous labels; we take the ogbn-arXiv graph and train GNNs with minibatching with on the original ogbn-arXiv subject area labels (that are homophilous, see Table 5), and also train GNNs with minibatching on the arXiv-year labels that we defined in this work (that are non-homophilous, see Table 2). For the minibatching methods, we use similar hyperparameters as in Appendix C.2 (Cluster-GCN uses 200 partitions and processes 5 parts at once, GraphSAINT-Node has a 10,000 node budget and GraphSAINT-RandWalk has 10,000 roots), which we call the "coarse" setting. In the so-called "fine" setting, we use smaller subgraph minibatches (Cluster-GCN uses 750 partitions and processes 5 parts at once, GraphSAINT-Node has a 2,500 node budget and GraphSAINT-RandWalk has 2,500 roots). Both the GCNJK and MixHop architectures are fixed to have two layers and 128 hidden dimensions.

Table 8 shows the results of these arXiv experiments on two different label sets. We see that performance degradation as measured by percent relative error in test accuracy is more substantial in the non-homophilous arXiv-year experiments for the GraphSAINT methods, while it is mostly comparable for the Cluster-GCN methods.

Graph minibatching could perform poorly in non-homophilous settings for a variety of reasons. It has been shown both theoretically and empirically that higher-order information from more than one-hop neighbors are important for classification in certain non-homophilous settings [82, 3]. One reason why graph minibatching may have issues here is that it is difficult to preserve higher-order neighborhood structure when minibatching on graphs. For instance, suppose we minibatch a graph with $n$ nodes by taking an induced subgraph on $n/2$ randomly selected nodes. Then for a node $u$ in the subgraph, each one hop path $u \to v$ has probability $1/2$ of being in the subgraph, whereas each two-hop path $u \to w \to v$ has probability $1/4$ of being in the subgraph, since both $w$ and $v$ must be sampled.

Finally, even if the percent relative error of minibatching methods is similar in homophilous vs. non-homophilous graphs, the performance degradation would generally be more detrimental in non-homophilous graphs. This is because the gap between GNNs and methods that do not use the graph topology like MLPs are lower in many non-homophilous settings. Since MLPs do not face much performance degradation when trained with simple i.i.d. node minibatching, the gap between GNNs and MLPs is even lower in minibatched settings. Indeed, we see that MLPs perform on par with or outperform many GNNs in the minibatched setting in Table 4.

Table 9: Minibatching results when using GCN as a base model. The setup is the same as in Section 5.3. LINKX results are provided as reference.

| | Penn94 | pokec † | arXiv-year | snap-patents † | genius | twitch-gamers † | wiki † |
|---|---|---|---|---|---|---|---|
| GCN-Cluster | 70.24±0.24 | 67.33±0.21 | 45.80±0.22 | 38.53±0.08 | 82.12±0.40 | 60.71±0.15 | (T) |
| GCN-SAINT-Node | 69.41±0.55 | 59.91±0.08 | 44.92±0.23 | 27.16±0.16 | 80.95±0.09 | 59.03±0.19 | 42.59±0.09 |
| GCN-SAINT-RW | 69.79±0.31 | 62.50±0.18 | 48.07±0.21 | 32.75±0.08 | 80.98±0.25 | 59.52±0.08 | 44.22±0.18 |
| LINKX Minibatch | 84.50±0.65 | 81.27±0.38 | 53.74±0.27 | 60.27±0.29 | 85.81±0.10 | 65.84±0.19 | 59.80±0.41 |

**GCN results.** In Section 5.3, we use GCNJK and MixHop as base models for minibatching evaluation. For completeness, we also include results for using GCN as a base model in Table 5.3. The results are qualitatively the same, though GCN generally performs worse than the other two GNNs.

## C.3 Experiments on Prior Datasets

Although the Pei et al. [58] datasets suffer issues with evaluation of graph learning methods as discussed in the main paper, we test LINKX on these datasets for comparison. We use the 10 fixed splits of [58], directly reporting results from papers where they also use the official splits, and re-running methods when this was not the case. In particular, GPR-GNN used larger 60-20-20 random splits, while GCNII did not evaluate on Actor and Squirrel. Thus, we re-run GPR-GNN and GCNII using the hyperparameters of Section B.1.

Table 10: Comparison of non-homophilous methods on datasets of Pei et al. [58] (collected by [61, 66, 48]). † represents re-run result, if unofficial dataset splits were used in their method paper, or the paper did not evaluate their method on the specific dataset. Best results up to a standard deviation are highlighted. Geom-GCN and GCNII did not report standard deviation information.

| # Nodes | Texas 183 | Wisconsin 251 | Actor 7,600 | Squirrel 5,201 | Chameleon 2,277 | Cornell 183 |
|---|---|---|---|---|---|---|
| H$_2$GCN-1 | 84.86 ± 6.77 | 86.67 ± 4.69 | 35.86 ± 1.03 | 36.42 ± 1.89 | 57.11 ± 1.58 | 82.16 ± 4.80 |
| H$_2$GCN-2 | 82.16 ± 5.28 | 85.88 ± 4.22 | 35.62 ± 1.30 | 37.90 ± 2.02 | 59.39 ± 1.98 | 82.16 ± 6.00 |
| MixHop | 77.84 ± 7.73 | 75.88 ± 4.90 | 32.22 ± 2.34 | 43.80 ± 1.48 | 60.50 ± 2.53 | 73.51 ± 6.34 |
| GPR-GNN | 76.22 ± 10.19† | 75.69 ± 6.59† | 33.12 ± 0.57† | 54.35 ± 0.87† | 62.85 ± 2.90† | 68.65 ± 9.86† |
| GCNII | 77.84 | 81.57 | 34.36 ± 0.77† | 56.63 ± 1.17† | 62.481 | 76.46 |
| Geom-GCN-I | 57.58 | 58.24 | 29.09 | 33.32 | 60.31 | 56.76 |
| Geom-GCN-P | 67.57 | 64.12 | 31.63 | 38.14 | 60.90 | 60.81 |
| Geom-GCN-S | 59.73 | 56.67 | 30.30 | 36.24 | 59.96 | 55.68 |
| LINKX | 74.60 ± 8.37 | 75.49 ± 5.72 | 36.10 ± 1.55 | 61.81 ± 1.80 | 68.42 ± 1.38 | 77.84 ± 5.81 |

For these experiments, we used the following hyper-parameter grid for LINKX: MLP$_{\mathbf{A}} \in \{1, 2\}$, MLP$_{\mathbf{X}} \in \{1, 2\}$, hidden channels $\in \{64, 128, 256, 512\}$, number of MLP$_f$ layers $\in \{1, 2, 3, 4\}$ learning rate in $\{0.05, 0.01, 0.002\}$ and dropout $\in \{0.0, 0.5\}$.

LINKX performs well on datasets with thousands of nodes, despite being primarily a scalable method, only falling short on the tiniest of datasets. Overall, evaluation on these datasets is very noisy, highly dependent on hyperparameter grid selection, and few conclusions can be drawn on the relative performance of different methods in non-homophily more broadly. In particular, note that methods that do well on these datasets do not necessarily do well on our large, non-homophilous datasets. For example, while MixHop is the best non-homophilous GNN on the datasets we present in this paper, it does poorly on the datasets of Pei et al. [58]; in contrast, $H_2$GCN achieves excellent performance here, but its design choices make it run out of memory on even medium-sized datasets.

### C.4 Experiments on Homophilous datasets

Though LINKX was not designed to do well on small homophilous datasets, we also report LINKX on the homophilous datasets of Cora, Citeseer and Pubmed [74], with the standard 48/32/20 training, validation, and test proportions. We use the same hyper-parameter grid for LINKX as in Section C.3.

Table 11: Comparison of on homophilous datasets. Results other than LINKX reported from [82]. Best three results per dataset are highlighted.

| # Nodes | CiteSeer 3327 | PubMed 19717 | Cora 2708 |
|---|---|---|---|
| $H_2$GCN-1 | $77.07 \pm 1.64$ | $89.40 \pm 0.34$ | $86.92 \pm 1.37$ |
| $H_2$GCN-2 | $76.88 \pm 1.77$ | $89.59 \pm 0.33$ | $87.81 \pm 1.35$ |
| MixHop | $76.26 \pm 1.33$ | $85.31 \pm 0.61$ | $87.61 \pm 0.85$ |
| GCN | $76.68 \pm 1.64$ | $87.38 \pm 0.66$ | $87.28 \pm 1.26$ |
| GAT | $75.46 \pm 1.72$ | $84.68 \pm 0.44$ | $82.68 \pm 1.80$ |
| LINKX | $73.19 \pm 0.99$ | $87.86 \pm 0.77$ | $84.64 \pm 1.13$ |

## D   Further dataset details

### D.1   Licenses

In this section, we note the licenses of the datasets we collect:

- wiki: Wikipedia is licensed under Creative Commons Attribution-ShareAlike 3.0 Unported License and the GNU Free Documentation License, unversioned, with no invariant sections, front-cover texts, or back-cover texts.

- Penn94: The Facebook 100 datasets are available online (https://archive.org/details/oxford-2005-facebook-matrix), and to the best of our knowledge were not released with a license, though the corresponding paper has an arXiv non-exclusive license to distribute. Upon release, there were privacy concerns [83], as the data release may not have respected certain privacy settings, and the initial release of the data included unique identifiers. We use a version that does not include the unique identifiers, but of course may still be subject to deanonymization attacks. While we do not add any sensitive information to the dataset, we acknowledge that deanonymization is possible, though our work does not directly contribute to deanonymization risks.

- Pokec: We retrieved the data from SNAP; the original source of the data is [40]. To the best of our knowledge, the data was not released with a license. This is a social network, so there may be privacy concerns with the user data. Still, we include it as a suitable benchmark that has been previously used, as the dataset is large and has interesting feature information. We only provide numerical values and do not provide any of the raw text in the dataset.

- arxiv-year: The dataset is licensed under ODC-BY. We originally downloaded the data from the Open Graph Benchmark [31], and the data is a subset of the Microsoft Academic Graph [70]

- snap-patents: The data was originally publically released by NBER in 2001 in a working paper by [27]. To the best of our knowledge, the dataset was not released with a license.

- genius: The dataset is open-sourced by the authors with no attached license. It was originally introduced in a conference paper [43]. While this is a social network and thus may

face privacy concerns, we believe that the task of predicting undesired nodes can be very beneficial to society, so we benchmark methods on it. We only provide numerical values, and omit raw text information that has been previously released in the dataset.

- twitch-gamer: The dataset is open-sourced by the authors in a repository with the MIT license. It was originally introduced in an academic paper [46]. This is also a social network that may face privacy concerns, but the prediction task of detecting situationally undesired nodes may be socially beneficial, so we benchmark methods on it. There is no raw text in the dataset.

## D.2  Dataset Properties

**Penn94** [67] is a friendship network from the Facebook 100 networks of university students from 2005, where nodes represent students. Each node is labeled with the reported gender of the user. The node features are major, second major/minor, dorm/house, year, and high school.

**Pokec** [41] is the friendship graph of a Slovak online social network, where nodes are users and edges are directed friendship relations. Nodes are labeled with reported gender. We derive node features from profile information, such as geographical region, registration time, and age.

**arXiv-year** [31] is the ogbn-arXiv network with different labels. Our contribution is to set the class labels to be the year that the paper is posted, instead of paper subject area. The nodes are arXiv papers, and directed edges connect a paper to other papers that it cites. The node features are averaged word2vec token features of both the title and abstract of the paper. The five classes are chosen by partitioning the posting dates so that class ratios are approximately balanced.

**snap-patents** [42, 41] is a dataset of utility patents in the US. Each node is a patent, and edges connect patents that cite each other. Node features are derived from patent metadata. Our contribution is to set the task to predict the time at which a patent was granted, resulting in five classes.

**genius** [43] is a subset of the social network on genius.com — a site for crowdsourced annotations of song lyrics. Nodes are users, and edges connect users that follow each other on the site. This social network has not been used for node classification in the literature, so we define the task of predicting certain marks on the accounts. About 20% of users in the dataset are marked "gone" on the site, which appears to often include spam users. Thus, we predict whether nodes are marked. The node features are user usage attributes like the Genius assigned expertise score, counts of contributions, and roles held by the user.

**twitch-gamers** [60] is a connected undirected graph of relationships between accounts on the streaming platform Twitch. Each node is a Twitch account, and edges exist between accounts that are mutual followers. The node features include number of views, creation and update dates, language, life time, and whether the account is dead. The binary classification task is to predict whether the channel has explicit content.

**wiki** is a dataset of Wikipedia articles, where nodes represent pages and edges represent links between them. We collect this new dataset, with a process that we describe further in Appendix D.3. Node features are constructed using averaged title and abstract GloVe embeddings [59]. Labels represent total page views over 60 days, which are partitioned into quintiles to make five classes.

## D.3  Wiki collection details

Here, we detail the process of crawling and cleaning the wiki dataset. We generated the graph using a breadth-first search, where we started from the Wikipedia page on Hilbert Spaces, then proceeded to visit all its neighbors, and so forth. For each Wikipedia page visited, we used the MediaWiki web service API to get a list of pages linked to by the given page — this forms the directed edges of the graph. In each API query, we also received the number of page views per day in the past 60 days. We crawled these articles throughout April and May of 2021. To convert this into discrete labels, we used an even quantile function, which set view boundaries to make the number of nodes in each class as even as possible. The output of this function formed the labels of the graph.

For the node features, we formed 300 dimensional Wikipedia Glove vectors [59] for each word in the title and abstract, then averaged the word vectors in the title and abstract, thus resulting in 600 dimensional feature vectors for each node. For words not found in the Glove dictionary, we used the

zero vector. This procedure was modeled on the construction of the ogbn-arxiv dataset by [31]. In particular, we avoided a one-hot vector because of the vast dimensionality that would be required. Finally, to clean the dataset, we pruned edges if either of the nodes that it spanned where not in our subset of wikipedia articles. Ultimately, we decided to stop collection at approximately one third of English wikipedia due to limitations of computational resources and time. Already, it is not possible to run full batch experiments on the wiki dataset, while requiring over 80GB of CPU RAM for some minibatching techniques. As such, we believe that the full wikipedia dataset would have been too large and unwieldy to use as an evaluation dataset for many research labs, and our dataset is at a good size that may hopefully provide utility to many researchers.