# OpenReview forum: "Large Scale Learning on Non-Homophilous Graphs: New Benchmarks and Strong Simple Methods"
_NeurIPS.cc/2021/Conference — NeurIPS 2021 Poster_

### Official Review · Reviewer_A2Hj · 2021-07-06

**Rating:** 6
**Confidence:** 3

**Summary:**

The paper studies the performance of graph methods on heterophilous datasets. In particular, they propose several new graph datasets, a method for such datasets, and an evaluation of many graph methods on these new datasets. The paper is written in a clear and thorough manner, there is an exhaustive list of references and introduction makes a good job of motivating the problems studied in the paper.

**Limitations And Societal Impact:**

As discussed above, one of the consequences of proposing real-world social networks and corresponding machine learning tasks of predicting properties of individuals (such as their gender) might be the misuse of the trained models on such datasets as well as promoting other works to propose datasets with sensitive information. As the field of GNNs mature, I can easily see how end users in the future use these models and datasets for the applications that treat unfairly certain groups of people (e.g. in credit scoring, institution admission, etc.).

**Main Review:**

The paper's contributions can be split into three parts: new datasets, new method, and empirical comparison.

1. I have strong concerns about the datasets and their possible misuse in the future. In particular, 4 of the proposed datasets are social networks or alike, which could be used to train machine learning models for assessing properties of individuals or the communities involved in the network. Besides, the tasks of predicting properties such as gender could lead to the situation when the trained models discriminate certain groups of people. As one of the main paper's contributions are the new datasets, I am worried that such datasets could be used in many other future related works that would eventually lead to unfair treatment of people by the machine learning models. Creating a precedent in this work, could also lead other works to include datasets that mistreat people's interests. I think there is [a related track](https://neurips.cc/Conferences/2021/CallForDatasetsBenchmarks) on benchmark and datasets, where this work can get more support and feedback on the validity of the proposed datasets.

2. The authors propose a new method called LINKX, which builds on a well-known method LINK. However, the previous LINK method was not used with node features and the authors propose to combine LINK embeddings with MLP embeddings on the node features. This allows the algorithm to scale easily to large datasets such as Wiki, whereas GNNs need more complex sampling solutions such as graphSAINT.

3. The empirical comparison is done nicely across the proposed datasets and many graph models including vanilla GNNs, heterophilous GNNs, MLPs, LP, and others. However, I would like to see the performance of LINKX on other datasets, homophilous and heterphilous (see [here](https://arxiv.org/abs/2102.06462) for example) for the fair comparison.

After rebuttal: Given a significant discussion between authors, reviewers, and ethics committee I tend to increase the score from 4 to 6. I still believe this paper would make a better fit to the dataset track of NeurIPS 2021.

**Needs Ethics Review:**

Yes

**Time Spent Reviewing:**

20

---

> ### Author Response · Authors · 2021-08-06
> **Author Response to Reviewer 3 (A2Hj) [Part 1]**
>
> We thank the reviewer for carefully considering the ethical implications of Machine Learning research, and understand the source of the reviewer’s concerns. Of course, we fully intend on reducing any potential for negative social consequences of our work. We look forward to the advice of the expert ethics reviewers, and we are prepared to take the necessary actions to reduce the potential for negative impacts of our work. We already briefly touch on some of these ethical issues in Section 6 of the main paper, but we promise to add to this in future versions of the paper with information from the discussions of this reviewing process.
>
> We would also like to clarify certain parts of our work and add useful information for further discussion. First, while the reviewer notes that 4/7 of our datasets are social networks, we note that 2 of these datasets are used for applications that have significant potential for positive social impact: detecting spam users on Genius, and detecting presence of mature content in twitch-gamers. Thus, we focus on the 2 datasets (Penn94 and pokec) for which we do reported gender prediction. If deemed necessary, we would be willing to discuss removing these 2 datasets from our paper and evaluation framework.
>
> Next, we review some prior work related to ours. Although the existence of such precedents clearly does not imply that our work may not have negative consequences, we would like to emphasize that we are not completely introducing new tasks from scratch, as these tasks come from established literature in computer science and social network analysis.
>
> * **Prior work introducing datasets:** As we detail in the paper, the only new dataset that we introduce is wiki — all other datasets have been studied in computer science previously (see citations in Section 3.2). For those previously appearing datasets, we only either define node features or new classification tasks. The social networks genius and twitch-gamers were introduced in detailed dataset-focused papers [Lim & Benson 21] [Rozemberczki & Sarkar 21].  For the gender prediction tasks, we introduce node features for Penn94 and Pokec.
>
> * **Prior work on gender / attribute prediction:** There has been a large body of work on the gender prediction task for the exact same prior datasets mentioned above. In machine learning and data science, gender prediction in Facebook100 graphs (the set of social networks that Penn94 is part of) has been extensively studied in a Nature paper [Altenburger & Ugander 18], and is a major focus of methods papers [Chin et al. 19] [Chen et al. 20] and an evaluation paper [Altenburger & Ugander 21]. Gender prediction in Pokec has been done in methods paper [Peel 17] [Kumar et al. 20]. Attribute prediction of users in social networks has been widely studied in the graph machine learning literature [Bhagat et al. 11] [Perozzi et al. 14].
>
> Given the prior work on these datasets, we stress that **we hide existing sensitive information and do not introduce new sensitive information**. Wiki, the only 1 out of our 7 proposed datasets that is not from previously published work, does not include user information. Moreover, in our evaluation framework, we do not provide any text in the datasets — just numerical data representing node features, graph edges, and node labels. We are not aware of sensitive information in these previously used datasets, though of course there is always the chance of deanonymization [Narayanan & Shmatikov 08]. Still, any actor who desires to deanonymize this data would not derive any benefit from our work, as the original sources of data provide more information than our evaluation framework.
>
> We thank the reviewer for taking the time to carefully consider the consequences of Machine Learning research, which too often has been under-appreciated. We fully acknowledge the reasons for thoroughly evaluating potential negative impacts of our work. We welcome further elaboration from the reviewer on specific areas of concern, and look forward to a fruitful discussion with the ethics reviewers.
>
> **References:**
> [Lim & Benson 21] Lim, D. and Benson, A.R., 2021, May. Expertise and Dynamics within Crowdsourced Musical Knowledge Curation: A Case Study of the Genius Platform. In Proceedings of the International AAAI Conference on Web and Social Media (ICWSM).
> [Rozemberczki & Sarkar 21] Rozemberczki, B. and Sarkar, R., 2021. Twitch Gamers: a Dataset for Evaluating Proximity Preserving and Structural Role-based Node Embeddings. arXiv preprint arXiv:2101.03091.
> [Altenburger & Ugander 18] Altenburger, K.M. and Ugander, J., 2018. Monophily in social networks introduces similarity among friends-of-friends. Nature human behaviour
> [Chin et al. 19] Chin, A., Chen, Y., M. Altenburger, K. and Ugander, J., 2019, May. Decoupled smoothing on graphs. In The World Wide Web Conference (WWW).
> [Chen et al. 20] Chen, Y., Tor, B., Augustine, E. and Getoor, L., 2020, January. Decoupled Smoothing in Probabilistic Soft Logic. In International Workshop on Mining and Learning with Graphs (MLG).
> [Altenburger & Ugander 21] Altenburger, K.M. and Ugander, J., 2021, May. Which Node Attribute Prediction Task Are We Solving? Within-Network, Across-Network, or Across-Layer Tasks. In Proceedings of the International AAAI Conference on Web and Social Media (ICWSM).
> [Peel 17] Peel, L., 2017, June. Graph-based semi-supervised learning for relational networks. In Proceedings of the 2017 SIAM International Conference on Data Mining. (SDM)
> [Kumar et al. 20] Kumar P, K., Langton, P. and Gatterbauer, W., 2020, June. Factorized Graph Representations for Semi-Supervised Learning from Sparse Data. In Proceedings of the 2020 ACM SIGMOD International Conference on Management of Data.
> [Bhagat et al. 11] Bhagat, S., Cormode, G. and Muthukrishnan, S., 2011. Node classification in social networks. In Social network data analytics (pp. 115-148). Springer, Boston, MA.
> [Perozzi et al. 14] Perozzi, B., Al-Rfou, R. and Skiena, S., 2014, August. Deepwalk: Online learning of social representations. In Proceedings of the 20th ACM SIGKDD international conference on Knowledge discovery and data mining
> [Narayanan & Shmatikov 08] Narayanan, A. and Shmatikov, V., 2008, May. Robust de-anonymization of large sparse datasets. In 2008 IEEE Symposium on Security and Privacy

---

> > ### Author Response · Authors · 2021-08-09
> > **Author Response to Reviewer 3 (A2Hj) [Part 2]**
> >
> > > “However, I would like to see the performance of LINKX on other datasets, homophilous and heterophilous (see here for example) for the fair comparison.”
> >
> > First, we note that LINKX results for all of the previously used standard heterophilous graphs in the Yan et al. paper are already included (see our Appendix Table 9).
> >
> > For the homophilous citation graphs, here is the LINKX performance. We follow the Yan et al. paper in using 48%/32%/20% train/validation/test proportions. We use the same hyperparameter grid for LINKX as in our Appendix section C.3:
> >
> > | Dataset | Mean accuracy ± stdev |
> > | ---| --- |
> > | Cora | 84.64 ± 1.13 |
> > | CiteSeer | 73.19 ± 0.99  |
> > | PubMed | 87.86 ± 0.77|
> >
> > We will add these results to a future version of the paper.

---

### Official Review · Reviewer_coWt · 2021-07-17

**Rating:** 7
**Confidence:** 5

**Summary:**

This paper proposes 7 new large heterophily graph datasets (up to 2 million nodes) and also provides a simple, scalable yet effective baseline LINKX which combines the information from topology and node features.
The paper extensively evaluates the datasets on various baselines, including typical GNNs and GNNs specifically designed for heterophily graphs. They find that many baselines cannot compete with the simple baseline LINKX and some of them run out of memory.
It also points out the minibatching problem and provides thorough results for fullbatch and minibatch training.


**Limitations And Societal Impact:**

I have a few questions regarding the new datasets.
1. From Table 3, it seems that GCN, in general, performs well than MLP in your heterophily datasets. It means, unlike the experiments did in small heterophily datasets, GCN can utilize graph structures in the new datasets. In the H2GCN paper, it mentions a similar situation for chameleon and squirrel datasets and points out that it is because those two datasets contain many nodes that share common neighbors. I am wondering for the heterophily datasets you provided, does this phenomenon occur in your datasets: nodes of same label share less connections but they share much more common neighbors than nodes from different classes. Can you also provide information on the 2-hop heterophily level, that is the portion of nodes from the same class in the 2-hop neighborhood but not the 1-hop neighborhood?
2. Recent works also find degree distribution affect the performance of node classification task. I am wondering if you can also provide the information on degree distribution on those datasets?

**Main Review:**

I appreciate the authors' effort in proposing large-scale heterophily graph datasets and along with a strong baseline LINKX.
Recently, many researchers have looked at the performances of GNNs on heterophily graphs but most of them are testing their models on very small heterophily datasets. So this work will potentially have a great impact on the community.
The authors not only provide various baselines (including typical GNNs and GNNs specifically designed for heterophily graphs) but also provide thorough results for fullbatch and minibatch training.

This paper is well written and easy to follow.

**Time Spent Reviewing:**

1hr

---

> ### Author Response · Authors · 2021-08-06
> **Author Response to Reviewer 2 (coWt)**
>
> Thank you for the positive comments and interesting considerations.
>
> > “Can you also provide information on the 2-hop heterophily level, that is the portion of nodes from the same class in the 2-hop neighborhood but not the 1-hop neighborhood?”
>
> This is an expensive operation, so we approximate it. We estimate the node homophily of the graph, where the neighborhood of each node is defined to be the nodes that are exactly two hops away (so no one-hop neighbors are included). In other words, the two-hop homophily measure we consider is:
>
> $$ \frac{1}{|V|} \sum_{v \in V} \frac{\text{Number of exact two-hop neighbors with same class as v}}{\text{Number of v’s exact two-hop neighbors}}. $$
>
> We estimate this by only summing over a random sample of $k$ nodes, and then replacing the normalization of $\frac{1}{|V|}$ by $\frac{1}{k}$. Note that for each node $v$ in this sample, the inner sum is still using the two-hop neighborhoods of the original graph. Here are computed numbers for $k=500$ (left is this two-hop measure, right is standard node homophily in equation (7) ):
>
> |Dataset | Two-hop | One-hop |
> | --- | --- | --- |
> |Penn94 | .474 | .483 |
> |pokec | .611 | .428 |
> |arXiv-year | .341 | .289 |
> |snap-patents | .330 | .221 |
> |genius | .749 | .508 |
> |twitch-gamers | .513 | .556 |
>
> Thus, the two-hop homophily is mostly similar to or somewhat higher than the standard one-hop node homophily. This is an interesting property that we will include in our paper revision.
>
> > “Recent works also find degree distribution affects the performance of node classification task. I am wondering if you can also provide the information on degree distribution on those datasets?”
>
> Here is an [anonymous imgur link](https://imgur.com/a/m7Fs3TQ) to figures showing the degree distributions for our datasets. We will include these plots in the future versions of the paper.

---

> > ### Comment · Reviewer_coWt · 2021-08-30
> > **Feedback to your response**
> >
> > I have no further questions and thanks for authors' effort in the rebuttal.

---

### Official Review · Reviewer_yhQP · 2021-07-30

**Rating:** 7
**Confidence:** 4

**Summary:**

The paper collects graph datasets that are not as homophile as the regular benchmarks for GNNs. Furthermore, it shows that a rather simple new baseline achieves quite good results compared to much more involved models / the SOTA. The paper presents important contributions to the community, is well and thoroughly written, related work is covered adequately, the experiments are overall thorough, and I have few to criticise. Altogether, I suggest to accept the paper.


**Ethical Concerns:**

None.

**Limitations And Societal Impact:**

Yes.

**Main Review:**

As outlined above, I have few to critisize, I summarise the most important contributions below.

- The collected (and sometimes adapted) datasets represent a useful set to move research in the field forward.

- The paper actually introduces a new metric to measure homophily which is quite interesting since it overcomes issues with existing ones. I would strongly suggest to move parts of all the explanations in the appendix into the paper, especially the descriptions illustrating the latter issues. I personally consider this metric contribution at least as important as the new model.

- The proposed approach is simple but effective which is quite nice, and the experiments over wiki show that such a solution was needed.

- The appendix contains much additional information and insights.

-------------------------------------------------------------
Questions & Smaller Comments
-------------------------------------------------------------

I wonder if the prediction of the year for the arxiv and patent data is a realistic task. Why would someone care about this in particular, and is it even possible to do this (ie do we expect enough variability in the data to derive such conclusions)?

**Time Spent Reviewing:**

3

---

> ### Author Response · Authors · 2021-08-06
> **Author Response to Reviewer 1 (yhQP)**
>
> We would like to thank you for the insightful remarks on our work.
>
> > "The paper actually introduces a new metric to measure homophily which is quite interesting since it overcomes issues with existing ones. I would strongly suggest to move parts of all the explanations in the appendix into the paper, especially the descriptions illustrating the latter issues. I personally consider this metric contribution at least as important as the new model.”
>
> Thank you for your positive comments on our homophily measure. We initially had this in the main paper, but moved it to the appendix for sake of space. With the additional space in the camera-ready, we will move much of this content to the main paper — especially the limitations of the previous measures and the formulation of our measure.
>
> > “I wonder if the prediction of the year for the arxiv and patent data is a realistic task. Why would someone care about this in particular, and is it even possible to do this (ie do we expect enough variability in the data to derive such conclusions)?”
>
> For the most part, this is a test task that has interesting non-homophilous structure (see the triangular compatibility matrices in Figure 3). Given that we achieve above 55% accuracy over 5 classes on these datasets, there is certainly a lot of structure in these datasets that can be used for successfully predicting years of nodes. Even LINK, which does not use node features, can predict node year well with just graph structure.
>
> We are not aware of currently popular applications, although we have seen year prediction tasks in past work (on smaller graphs) [Peel 17] [Kumar et al. 20]. There may also be possible applications in dating of historical documents or other archaeological tasks [Klassen et al. 18].
>
> **References:**
> [Peel 17] Peel, L., 2017, June. Graph-based semi-supervised learning for relational networks. In Proceedings of the 2017 SIAM International Conference on Data Mining. (SDM)
> [Kumar et al. 20] Kumar P, K., Langton, P. and Gatterbauer, W., 2020, June. Factorized Graph Representations for Semi-Supervised Learning from Sparse Data. In Proceedings of the 2020 ACM SIGMOD International Conference on Management of Data.
> [Klassen et al. 18] Klassen, S., Weed, J. and Evans, D., 2018. Semi-supervised machine learning approaches for predicting the chronology of archaeological sites: A case study of temples from medieval Angkor, Cambodia. PloS one.

---

> > ### Comment · Reviewer_yhQP · 2021-08-20
> > **Response**
> >
> > I acknowledge that I have read the response as well as the other reviews and responses.
> >
> > The ethical concerns are an important point to consider. However, I feel that this should not be a reason for rejecting this paper, the data is not too uncommon. This discussion needs to be started on a global level, and I would hope that the community discusses and comes up with guidelines before it is the EU. It would be great if people with greater influence such as the conference and area chairs would consider moving this discussion forward.

---

### Review · Ethics_Reviewer_EeVj · 2021-08-12

**Recommendation:**

Yes, it is possible to address the concerns in the current version of the paper. See my suggestions above.

**Ethical Issues:**

Yes

**Ethics Review:**

Reviewer 3 indeed brought up relevant ethical issues related to this paper, in particular concerning the misuse of the datasets to discriminate against certain groups of people. As the authors acknowledge, such datasets have also been used in the past in machine learning papers. Thus, it is important to start establishing a precedent concerning the "misuse" of datasets and methods. The omission of sensitive, identifying information is a low benchmark. As the authors acknowledge in section 6, "Negative social consequences may occur if these methods are deployed for nefarious purposes; for example, node classification may be used to infer users’ gender, sexual orientation, and political beliefs, and thus cause users to face discrimination and unfair treatment." It is important to acknowledge why these datasets or methods could be deployed for nefarious purposes. Which individuals, groups, or institutions might have access to these data, due to their positions in society? Why might they want access to these data? Why might these datasets get used for malicious purposes? While it might be beyond the scope of this paper to address these issues directly, it is important for the discussion section of this paper to acknowledge an awareness of how this paper would sit within networks of knowledge and power.

---

> ### Author Response · Authors · 2021-08-21
> **Author Response to Ethics Reviewer EeVj**
>
> We thank the reviewer for these insights. We will incorporate necessary discussions into our main paper, as outlined in the general ethics comment.
>
> > “Which individuals, groups, or institutions might have access to these data, due to their positions in society?”
>
> See Paragraph 2 of the Broader Impacts section in the general comment.
>
> > “Why might they want access to these data?”
>
> See Paragraph 2 of the Broader Impacts section in the general comment.
>
> > “Why might these datasets get used for malicious purposes?”
>
> See Paragraph 2 of the Broader Impacts section in the general comment. We have also added possible negative consequences when the datasets are not purposefully used maliciously (see Paragraph 3 of the Broader Impacts).

---

> > ### Comment · Ethics_Reviewer_EeVj · 2021-08-26
> > **Clarification**
> >
> > Thank you to the authors for addressing these reviews extensively. I would encourage the authors to reconsider using the term "bad actors" to refer to individuals, groups, or institutions that might have access to these data and use them for their own purposes. Using a term such as "bad actors" implies that such entities are somehow dispositionally nefarious, when that is hardly the case. I would encourage the authors to consider how particular entities that might have more power (e.g., access to capital, data, and other resources) might be more likely to act in their own interests. The authors point to how there might only be a few "bad actors" such as social media companies. However, these social media companies could do much damage, as we have seen with Facebook.
> >
> > In the ethics discussion, I would encourage the authors to remove the term "bad actors" and discuss these entities situationally rather than dispositionally.

---

> > > ### Author Response · Authors · 2021-08-27
> > > **Author Response to Clarification by EeVj**
> > >
> > > Thank you for bringing this up. We appreciate this perspective, and agree with the Ethics Reviewer's argument; entities using the data will not tend to be generically nefarious, but instead there may be concerns on situational use.
> > >
> > > We will indeed remove the term "bad actors". We will instead emphasize that different entities could situationally use data in ways that raise ethical concerns. Also, we agree that there is still potential for high damage from few entities, and we will make a disclaimer stating this in the text.

---

### Review · Ethics_Reviewer_RMnL · 2021-08-12

**Recommendation:**

yes, but it will require additional documentation in the appendix as well as discussion in the text

**Ethical Issues:**

Yes

**Ethics Review:**

The authors note
"Negative social consequences may occur if these methods are deployed for nefarious purposes; for
example, node classification may be used to infer users’ gender, sexual orientation, and political beliefs, and thus cause users to face discrimination and unfair treatment."
However they do not cite or discuss existing literature identifying such consequences. As reviewer A2Hj notes the paper should include a meaningful discussion of risks.

While the majority of the data sets they use are already publicly available and used in prior research, this doesn't eliminate the need to note ethical concerns with harvesting data that individuals have affirmatively shared or have unwittingly revealed through online activity for other purposes--or at least to point to existing literature that does so. Both data individuals share on online platforms, and the trace data generated by their activities, are not akin to data a researcher might collect through observations  in physical public places. Individuals often share information without attention to the potential audience, the permanance and discoverability. They are completely unaware of the trace data, and of course of the potential inferences that can be drawn from it given the increased power and sophistication of computational tools. And as the authors note there is increased risk of data sets being deanonymized. This is particularly true as new reconstruction attacks pave the way for reidentification attacks.

Emerging best practices require data sets to be accompanied by documentation that supports sound and ethical use.The addendum provides very limited information about the existing datasets and makes no effort to assess whether they are appropriate to use in benchmarking or other research. I believe at least one of the datasets "Penn94: The Facebook 100 datasets are available online, and to the best of our knowledge were not released with a license." was controversial when produced (raising privacy and other concerns) which is not mentioned. The new data set does not adhere to the emerging best practices wrt documentation.

---

> ### Author Response · Authors · 2021-08-21
> **Author Response to Ethics Reviewer RMnL**
>
> We thank the ethics reviewer for an insightful assessment of our current ethics discussion, and for adding important points that we have overlooked. We will incorporate these points into our paper as outlined in the general ethics comment.
>
> > “Both data individuals share on online platforms, and the trace data generated by their activities, are not akin to data a researcher might collect through observations in physical public places. Individuals often share information without attention to the potential audience, the permanance and discoverability. They are completely unaware of the trace data, and of course of the potential inferences that can be drawn”
>
> We have now addressed these important points, primarily in paragraph 4 of the new Broader Impacts section (see general comment).
>
> > “And as the authors note there is increased risk of data sets being deanonymized. This is particularly true as new reconstruction attacks pave the way for reidentification attacks.”
>
> See paragraph 2 of new Broader Impacts section in the general comment.
>
> > “Emerging best practices require data sets to be accompanied by documentation that supports sound and ethical use.The addendum provides very limited information about the existing datasets and makes no effort to assess whether they are appropriate to use in benchmarking or other research.”
>
> See revisions to Appendix D in the general comment.
>
> > “I believe at least one of the datasets "Penn94: The Facebook 100 datasets are available online, and to the best of our knowledge were not released with a license." was controversial when produced (raising privacy and other concerns) which is not mentioned”
>
> We thank the reviewer for pointing this out, as we were not aware of this. We add a discussion and a citation to relevant work on privacy concerns with the dataset in Appendix D in the general comment.
>
> > “The new data set does not adhere to the emerging best practices wrt documentation.”
>
> We will reorganize our Appendix Section D.1 on the documentation of our new wiki dataset into a Datasheet [Gebru et al. 18]. We will also add further documentation where we are lacking.

---

### Author Response · Authors · 2021-08-09
**General Response to All Reviewers**

Firstly, we would like to express our appreciation for the work that the reviewers and organizing members have put in, and thank all reviewers for their valuable feedback.

We highlight here some positive comments on our work:
* LINKX model: “simple but effective” (R1), “simple, scalable yet effective” (R2), “scale easily to large datasets” (R3).
* Dataset: “datasets represent a useful set to move research in the field forward” (R1), “will potentially have a great impact on the community” (R2).
* Evaluation: “experiments are overall thorough” (R1), “extensively evaluates the datasets on various baselines” “thorough results” (R2), “empirical comparison is done nicely” “many graph models” (R3).
* Writing: “well and thoroughly written” (R1), “well written and easy to follow” (R2), “written in a clear and thorough manner” (R3).

Review 3 raised potential ethical concerns and called for an ethics review. We added to this discussion and addressed some points in the comment below Review 3. Besides that, we welcome discussion and further input by the ethics reviewers.

Our revised paper will have some changes that are mentioned in the individual comments:
* We will move some of the content of Appendix A.1 to the main paper, as Reviewer 1 believes that this content is “interesting” and “important”.
* We will include two-hop node homophily statistics as in the comment under Review 2.
* We will include degree distribution plots for our datasets as in the comment under Review 2.
* We will include results of LINKX on some homophilous citation datasets as in the comment under Review 3.
* We will include additional discussions of ethical concerns, and possibly make other changes addressing ethical concerns brought up in this reviewing period.

Once again, we thank the reviewers for their time, and respond to individual comments with clarifications and additional results below. Addressing these concerns have strengthened our paper, and our fundamental results remain the same.

---

### Author Response · Authors · 2021-08-21
**General Ethics Review Comment**

We thank the ethics reviewers for adding their expertise into this fruitful discussion. We have taken their insights into account, and have done further consideration and research of our own into ethical concerns. Also, we thank reviewer yhQP for their recently-added input on the ethical concerns, and their positive assessment of our work. In this comment we give new additions to our paper that revise and substantially add to the discussion of societal impacts and the documentation of data in our submission.

**Broader impacts** (revision of Broader impacts subsection of section 6 in main paper).

Fundamental research in graph learning on non-homophilous graphs has the potential for positive societal benefit. As a major application, it enables malicious node detection techniques in social networks and transaction networks that are not fooled by fraudsters’ connections to legitimate users and customers. This is a widely studied task, and past works have noted that non-homophilous structures are present in many such networks [Breuer et al. 20] [Gatterbauer 14] [Pandit et al. 07]. We hope that this paper provides insight on the homophily limitations of existing scalable graph learning models and help researchers design scalable models that continue to work well in the non-homophilous regime, thus improving the quality of node classification on graphs more broadly. As our proposed datasets have diverse structures and our model performs well across all of these datasets, the potential for future application of our work to important non-homophilous tasks is high.

Nevertheless, our work could also have potential for different types of negative social consequences. Nefarious behavior by bad actors could be one source of such consequences. For example, node classification may be used to infer sensitive attributes such as gender [Altenburger & Ugander 18], sexual orientation [Jernigan & Mistree 09], and political beliefs [Lindamood et al. 09], which could be used by a bad actor to discriminate against users in several ways. Still, we expect that the bad actors that can make use of large-scale social networks for gender prediction as studied in our work are limited in number. Such bad actors would probably mostly consist of entities with access to large social network data such as social media companies or government actors with auxiliary networks [Narayanan & Shmatikov 09]. Their motivations may vary, but social media companies may want user information to better serve advertisements, and government actors may derive strategic advantages from user information. Smaller actors can perform certain attacks, but this may be made more difficult by resource requirements such as the need for certain external information [Narayanan & Shmatikov 09] or the ability to add nodes and edges before an anonymized version of a social network is released [Backstrom et al. 07]. Furthermore, bad actors could make use of deanonymization attacks [Hay et al. 10] [Narayanan & Shmatikov 08] [Narayanan & Shmatikov 09] to reveal user identities in supposedly anonymized datasets.

Also, accidental consequences and implicit biases are a potential issue, even if the applications of the graph machine learning algorithms are benign and intended to benefit society [Mehrabi et al. 21]. Performance of algorithms may vary substantially between intersectional subgroups of subjects — as in the case of vision-based gender predictors [Buolamwini & Gebru 18]. Thus, there may be disparate effects on different populations, including protected subgroups. Moreover, large datasets require computing resources so that projects can only be pursued by large entities at the possible expense of the individual and smaller research groups [Birhane et al. 21]. This is alleviated by the fact that our experiments are each run on one GPU, and hence have significantly less GPU computing requirements than much current deep learning research. Thus, smaller research groups and independent researchers should find our work beneficial, and should be able to build on it.

Finally, the nature of collection of online user information also comes with notable ethical concerns. Common notice-and-consent policies are often ineffective in actually protecting user privacy [Nissenbaum 11]. Indeed, users may not actually have much choice in using certain platforms or sharing data due to social or economic reasons. Also, users are generally unable to fully read and understand all of the different privacy policies that they come across, and may not understand the implications of having their data available for long periods of time to entities with powerful inference algorithms. Furthermore, people may rely on obscurity for privacy [Hartzog & Stutzman 13], but this assumption may be ignored in courts of law, and it may be directly broken when data leaks or is released in aggregated form. Overall, while we believe that our work will benefit machine learning research and enable positive applications, we must still be aware of possible negative consequences.


**Appendix D: Further Dataset Details** (revision of Appendix D)

In this section, we give licenses and further information on the datasets we use:

* wiki: Wikipedia is licensed under Creative Commons Attribution-ShareAlike 3.0 Unported License and the GNU Free Documentation License, unversioned, with no invariant sections, front-cover texts, or back-cover texts. Further information is given in Appendix D.1.
* Penn94: The Facebook 100 datasets are available online (https://archive.org/details/oxford-2005-facebook-matrix), and to the best of our knowledge were not released with a license, though the corresponding paper has an arXiv non-exclusive license to distribute. Upon release, there were privacy concerns [Zimmer 11], as the data release may not have respected certain privacy settings, and the initial release of the data included unique identifiers. We use a version that does not include the unique identifiers, but of course may still be subject to deanonymization attacks. While we do not add any sensitive information to the dataset, we acknowledge that deanonymization is possible, though our work does not directly contribute to deanonymization risks.
* Pokec: We retrieved the data from SNAP; the original source of the data is [35]. To the best of our knowledge, the data was not released with a license. This is a social network, so there may be privacy concerns with the user data. Still, we include it as a suitable benchmark that has been previously used, as the dataset is large and has interesting feature information. We only provide numerical values and do not provide any of the raw text in the dataset.
* arxiv-year: The dataset is licensed under ODC-BY. We originally downloaded the data from the Open Graph Benchmark [25], and the data is a subset of the Microsoft Academic Graph [Wang et al. 20]
* snap-patents: The data was originally publically released by NBER in 2001 in a working paper by Hall et al. [24]. To the best of our knowledge, the dataset was not released with a license.
* genius: The dataset is open-sourced by the authors with no attached license. It was originally introduced in a conference paper [36]. While this is a social network and thus may face privacy concerns, we believe that the task of predicting undesired nodes can be very beneficial to society, so we benchmark methods on it. We only provide numerical values, and omit raw text information that has been previously released in the dataset.
* twitch-gamer: The dataset is open-sourced by the authors in a repository with the MIT license. It was originally introduced in an academic paper [48]. This is also a social network that may face privacy concerns, but the prediction task of detecting situationally undesired nodes may be socially beneficial, so we benchmark methods on it. There is no raw text in the dataset.

---

> ### Author Response · Authors · 2021-08-21
> **References for General Ethics Review Comment**
>
> **References:**
>
> [Gebru et al. 18] Gebru, T., Morgenstern, J., Vecchione, B., Vaughan, J.W., Wallach, H., Daumé III, H. and Crawford, K., 2018. Datasheets for datasets. arXiv:1803.09010.
> [Buolamwini & Gebru 18] Buolamwini, J. and Gebru, T., 2018, January. Gender shades: Intersectional accuracy disparities in commercial gender classification. In Conference on fairness, accountability and transparency.
> [Zimmer 11] Zimmer, M.  Facebook Data of 1.2 Million Users from 2005 Released: Limited Exposure, but Very Problematic. Blog post. https://michaelzimmer.org/2011/02/15/facebook-data-of-1-2-million-users-from-2005-released/
> [Breuer et al. 20] Breuer, A., Eilat, R. and Weinsberg, U., 2020, April. Friend or faux: Graph-based early detection of fake accounts on social networks. In Proceedings of The Web Conference 2020.
> [Gatterbauer 14] Gatterbauer, W., 2014. Semi-supervised learning with heterophily. arXiv:1412.3100.
> [Pandit et al. 07] Pandit, S., Chau, D.H., Wang, S. and Faloutsos, C., 2007, May. Netprobe: a fast and scalable system for fraud detection in online auction networks. In WWW.
> [Birhane et al. 21] Birhane, A., Kalluri, P., Card, D., Agnew, W., Dotan, R. and Bao, M., 2021. The Values Encoded in Machine Learning Research. arXiv:2106.15590.
> [Hay et al. 10] Hay, M., Miklau, G., Jensen, D., Towsley, D. and Li, C., 2010. Resisting structural re-identification in anonymized social networks. The VLDB Journal.
> [Narayanan & Shmatikov 08] Narayanan, A. and Shmatikov, V., 2008, May. Robust de-anonymization of large sparse datasets. In 2008 IEEE Symposium on Security and Privacy
> [Narayanan & Shmatikov 09] Narayanan, A. and Shmatikov, V., 2009, May. De-anonymizing social networks. In 2009 30th IEEE symposium on security and privacy.
> [Backstrom et al. 07] Backstrom, L., Dwork, C. and Kleinberg, J., 2007, May. Wherefore art thou R3579X? Anonymized social networks, hidden patterns, and structural steganography. In WWW.
> [Nissenbaum 11] Nissenbaum, H., 2011. A contextual approach to privacy online. Daedalus.
> [Hartzog & Stutzman 13] Hartzog, W. and Stutzman, F., 2013. The case for online obscurity. Calif. L. Rev.
> [Mehrabi et al. 21] Mehrabi, N., Morstatter, F., Saxena, N., Lerman, K. and Galstyan, A., 2021. A survey on bias and fairness in machine learning. ACM Computing Surveys (CSUR).
> [Lindamood et al. 09] Lindamood, J., Heatherly, R., Kantarcioglu, M. and Thuraisingham, B., 2009, April. Inferring private information using social network data. In WWW.

---

### Decision · Program_Chairs · 2021-09-27

**Decision:**

Accept (Poster)

**Comment:**

The paper proposes a new bechmark for graph neural networks for non-homophilic graphs. This is an important aspect currently missing in existing datasets. The reviewers were generally positive about the paper, and most of the review comments raised ethics concerns which in the opinion of the AC have been properly addressed by the authors. The authors provided detailed responses in the rebuttal. We recommend acceptance.